# Lockdown-Related Disparities Experienced by People with Disabilities during the First Wave of the COVID-19 Pandemic: Scoping Review with Thematic Analysis

**DOI:** 10.3390/ijerph18126178

**Published:** 2021-06-08

**Authors:** Tiago S. Jesus, Sutanuka Bhattacharjya, Christina Papadimitriou, Yelena Bogdanova, Jacob Bentley, Juan Carlos Arango-Lasprilla, Sureshkumar Kamalakannan

**Affiliations:** 1Global Health and Tropical Medicine (GHTM) & WHO Collaborating Centre for Health Workforce Policy and Planning, Institute of Hygiene and Tropical Medicine, NOVA University of Lisbon, 1349-008 Lisbon, Portugal; 2Department of Occupational Therapy, College of Health & Rehabilitation Sciences: Sargent College, Boston University, Boston, MA 02215, USA; 3Department of Occupational Therapy, Byrdine F. Lewis College of Nursing and Health Professions, Georgia State University, Atlanta, GA 30303, USA; 4Departments of Interdisciplinary Health Sciences, and Sociology, School of Health Sciences, Oakland University, Rochester, MI 48309-4452, USA; 5Physical Medicine & Rehabilitation Service, Veterans Affairs Boston Healthcare System, Boston, MA 02130, USA; 6Department of Psychiatry, School of Medicine, Boston University, Boston, MA 02130, USA; 7Department of Clinical Psychology, Seattle Pacific University, Seattle, WA 98119, USA; 8Department of Physical Medicine & Rehabilitation, School of Medicine, John Hopkins University, Baltimore, MD 21205-2196, USA; 9IKERBASQUE, Basque Foundation for Science, 48009 Bilbao, Spain; 10Biocruces Bizkaia Health Research Institute, 48903 Barakaldo, Spain; 11Department of Cell Biology and Histology, University of the Basque Country UPV/EHU, 48940 Leioa, Spain; 12Public Health Foundation of India (PHFI), South Asia Centre for Disability Inclusive Development and Research (SACDIR), Indian Institute of Public Health, Hyderabad 500033, India

**Keywords:** COVID-19, SARS-CoV-2, health equity, social determinants of health, people with disabilities, public health, healthcare disparities, discrimination, stigma, social inclusion

## Abstract

People with disabilities may be disproportionally affected by the COVID-19 pandemic. We synthesize the literature on broader health and social impacts on people with disabilities arising from lockdown-related measures. Methods: Scoping review with thematic analysis. Up to mid-September 2020, seven scientific databases and three pre-print servers were searched to identify empirical or perspective papers addressing lockdown-related disparities experienced by people with disabilities. Snowballing searches and experts’ consultation also occurred. Two independent reviewers took eligibility decisions and performed data extractions. Results: Out of 1026 unique references, 85 addressed lockdown-related disparities experienced by people with disabilities. Ten primary and two central themes were identified: (1) Disrupted access to healthcare (other than for COVID-19); (2) Reduced physical activity leading to health and functional decline; (3) From physical distance and inactivity to social isolation and loneliness; (4) Disruption of personal assistance and community support networks; (5) Children with disabilities disproportionally affected by school closures; (6) Psychological consequences of disrupted routines, activities, and support; (7) Family and informal caregiver burden and stress; (8) Risks of maltreatment, violence, and self-harm; (9) Reduced employment and/or income exacerbating disparities; and (10) Digital divide in access to health, education, and support services. Lack of disability-inclusive response and emergency preparedness and structural, pre-pandemic disparities were the central themes. Conclusions: Lockdown-related measures to contain the COVID-19 pandemic can disproportionally affect people with disabilities with broader impact on their health and social grounds. Lack of disability-inclusive response and emergency preparedness and pre-pandemic disparities created structural disadvantages, exacerbated during the pandemic. Both structural disparities and their pandemic ramifications require the development and implementation of disability-inclusive public health and policy measures.

## Background

1

The coronavirus disease 2019 (COVID-19), caused by the severe acute respiratory syndrome coronavirus 2 (SARS-CoV-2) strain, emerged in late 2019, and since then has caused a global public health crisis of unusual proportions. Up to March 16, 2020, there were over 120 million cases and over 2.6 million deaths confirmed worldwide [1]. However, the impact of the COVID-19 pandemic, including the unintended effects of the measures to contain the pandemic (e.g., lockdowns), has not been equal across population groups [2-4].

People with disabilities include people who experience, at any given point in their lifespan, any mobility, intellectual, cognitive, developmental, or sensorial impairments which in interaction with environmental factors may hinder their daily functioning and social participation on an equal basis with others [5-7]. People with disabilities may be disproportionally affected by the COVID-19 pandemic. This disproportional impact entails greater risks of being infected (especially for people with disabilities living in residential or long-term care institutions) or greater risks of having severe health consequences once infected, including from unethical disadvantages in the access to life-saving treatments [8]. Furthermore, people with disabilities often require routine health and rehabilitative care (e.g., at home, outpatient) to maintain or recover their health and function. However, many of these services were considered non-essential, closed or functioning with important human and physical resources restrictions during initial lockdowns to contain the pandemic [9,10]. There are widely documented health status, health outcomes, and healthcare access disparities that people with disabilities have been experiencing for long time [11-13]. Yet, these forms of health disparities have been widened and exacerbated during the COVID-19 pandemic [8,14,15].

Additionally, people with disabilities are historically a socially vulnerable (not inherently vulnerable) and marginalized group whose social participation and welfare may be disproportionally affected by the COVID-19 pandemic. Lockdown, enforced quarantines, and other public health and policy measures aimed at containing the pandemic are often necessary. When not carefully planned, though, these measures can result in negative educational, occupational, and socio-economic consequences, which may hit harder the most socially vulnerable [4,16,17], including some people with disabilities [14,18]. Therefore, the impact of the pandemic can widen existing health and socio-economic disparities, if no protective actions focused on the most socially vulnerable are taken as counter-measures [2,3,18-22].

People with disabilities may disproportionally experience the negative health and socio-economic effects of lockdown-related measures. For example, people with disabilities as a group are more frequently resource-poor, have lower employment rates, additional health care and living costs, and less disposable income than non-disabled counterparts [23-26]. Furthermore, many people with disabilities often rely on formal and informal caregivers and social support networks to fulfill basic needs or live independently in the community; these supports may be disrupted during lockdowns or as result of quarantines of people with disabilities themselves or their caregivers’ [27,28]. Finally, tele-health or tele-schooling solutions were often not prepared to accommodate the needs of people with disabilities [29], including children with special education needs and their families, who may be especially affected by school closures [30,31]. Overall, people with disabilities regularly experience social participation disparities when they are denied, excluded, or deprived from an equal opportunity to pursue meaningful occupations, social roles, and social integration, when compared to people without disabilities [32,33]. In turn, these social participation disparities directly contribute to broader health disparities, as participating in meaningful occupations is a key determinant of human health and wellbeing [34,35].

In conclusion, people with disabilities can be especially vulnerable to negative effects of lockdown and other public health measures to contain the pandemic, especially when these measures are not disability-inclusive and not designed to prevent or mitigate any disproportionate impacts. Within this context, the purpose of this manuscript is to develop a scoping review of the literature on lockdown-related disparities that people with disabilities experienced in health, social participation, and socio-economic grounds during the initial stages of the COVID-19 pandemic, to inform disparities-reduction action from key stakeholders (e.g., policy makers, public health authorities, civil society).

## Methods

2

This paper uses a scoping review method with a thematic analysis as the analytical approach. The overall scoping review protocol, which covered a whole spectrum of vulnerabilities people with disabilities may be experiencing during the pandemic, was published a priori [5]. The disproportional health risks for or consequences of a COVID-19 infection are reported in another paper [8]. Here, we focus on the broader health and social disparities arising from lockdown-related measures. In reporting, we use the Preferred Reporting Items for Systematic reviews and Meta-Analyses extension for Scoping Reviews (PRISMA-ScR) [36].

### Eligibility Criteria

2.1

This scoping review included any peer-reviewed empirical, conceptual or perspective paper (e.g., editorials, commentaries) or preprint empirical studies explicitly addressing: (1) the COVID-19 disease or pandemic, (2) people with disabilities as a group, subgroup (e.g., based on impairment type or underlying diagnostic condition) or related individual circumstances; and (3) any vulnerabilities or disparities experienced by people with disabilities in terms of health and socio-economic impacts arising from lockdowns or other public health and policy measures to contain the pandemic.

Aligned with the content published in the open-access study protocol [5], the Supplementary Materials provides working definitions of people with disabilities and of vulnerability, including a text-box of illustrative examples of possible individual and multiple social vulnerabilities to the effects of the COVID-19 pandemic, as a means to inform eligibility decisions. For the scope of the paper, lockdown measures include: stay-at-home orders, curfews, in-country travelling restrictions, closures of schools and community supportive services, restriction in accessing welfare services, closures or restricted access to health services considered non-essential, and restrictions in visitor policies in residential and long-term care facilities. Related measures included quarantine or self-isolation periods, any policy or socio-economic measures to prevent or mitigate negative lockdown impacts, and could address the people with disabilities as well as any formal or informal caregiver they rely on. We had no geographic restrictions. Also, we searched for papers in six languages (i.e., English, French, Spanish, Greek, Russian, and Portuguese), yet only full-texts in English met all the eligibility criteria.

### Information Sources, Search, and Selection

2.2

Details of the information sources and search process can be found elsewhere [5,8]. In short, up to mid-September, seven scientific databases (Medline/PubMed, Web of Science, Scopus, AgeLine, PsycINFO, CINAHL, and ERIC) and three preprint servers (i.e., MedRxiv, SocArXiv, and PsyArXiv) were searched for to identify empirical or perspective papers meeting the eligibility criteria. During the initial stages of the COVID-19 pandemic, preprint databases have been hosting many studies that have not reached peer-reviewed publications yet [37]. This option can also help to avoid the exclusion of studies with negative results, which may be published less often or less rapidly. Before the data charting, we searched for the peer-reviewed version of the included preprints, and have replaced the record whenever found. The full search strategies for each database were outlined in the open-access protocol [5], and can be consulted at the Supplementary Materials. A snowballing search process (e.g., author tracking, referenced sources) and key-informants (i.e., members of the American Congress of Rehabilitation Medicine’s International Networking Group and Refugee Empowerment Task Force) had the opportunity to provide any additional references, supplied with a preliminary list of inclusions.

Key elements of the grey literature have been freely-accessible in a repositorium hosted by the United Nations (https://www.un.org/development/desa/disabilities/covid-19.html, accessed date: 15 December 2020), as identified by our initial searches. With this information already mapped, the review authors decided not to include this literature [8]. Two independent reviewers (SK, SB) made eligibility decisions in both titles-and-abstract screening and full-text assessments, after pilot screenings with over 80% agreements. Any’discrepancies were resolved through consensus or the leading author’s (TJ) input.

### Data Charting and Items

2.3

One author (SK) extracted formal data elements (e.g., publication type, sources, geographies addressed), following a pre-defined coding structure elaborated by members of the team, with a random 5% of the extractions verified by another (TJ). For the content, two independent reviewers (SK, SB) extracted text quotations on disproportional health and socio-economic impacts from the COVID-19 pandemic on people with disabilities regardless of having COVID-19 infection, i.e., as a result of lockdowns and other public health and policy measures to contain the pandemic. These independent extractions were later paired for the qualitative data synthesis, which was also informed by a brief synthesis of each paper developed by two reviewers independently. The Supplementary Materials provides the content of these extractions after being merged (i.e., presented as the combined extractions of both reviewers), as well as reviewers’ combined synthesis of each paper.

### Critical Appraisal

2.4

Quality assessments of methods were not performed as described in the study protocol [5], and common in scoping reviews [38-40].

### Synthesis of the Results

2.5

Simple descriptive statistics (e.g., counts, percentages) were computed to provide a summative description of the amount and range of the related literature, including publication type and source, country (or countries), or health conditions or impairments addressed. For the text quotations extracted, we have developed a thematic analysis [5,41,42], synthesizing the complex net of health, socio-economic or participation disparities experienced by people with disabilities as a result from lockdown-related measures. In this synthesis, we have developed a new interpretive schema and configuration, inclusive of both primary and central, overarching themes. To enable this type of qualitative synthesis, out of mixed-methods data coming from the scoping review, we applied an integrative, data-based convergent synthesis approach [43,44], where qualitative and quantitative evidence, as well as published perspectives, were analyzed using the same synthesis method. In this case, all forms of data were analyzed qualitatively and together under the same themes, provided that they addressed the same contents. Specifically, quantitative data were not numerically aggregated with data from other quantitative papers included, but analyzed in their meaning and extracted implications, always alongside qualitative data and published perspectives within the same themes [43,44]. In short, these different types of information were qualitatively combined within the same themes, in complement to (e.g., contributing to the interpretation of) one another.

Finally, to help the reader discern whether the reported material comes from empirical or perspectives papers, the synthesis explicitly reports a few study characteristics, populations, or any numerical findings when applicable. Finally, as described in the study protocol [5], we took a final consultation stage. Supplied with a preliminary version of the results and its discussion, members of the American Congress of Rehabilitation Medicine’s International Networking Group and Refugee Empowerment Task Force had the opportunity to comment and provide improvement suggestions over the preliminary results and their interpretation.

## Results

3


Figure 1 provides the flowchart of this review. Out of 1027 unique references, 85 are included in the final analysis, i.e., report findings or rationales for any disproportionate, lockdown-related health or social consequences for people with disabilities.


Table 1 shows how the papers that were analysed are distributed by publication type and source, by geographical focus, and by the health conditions or impairments addressed.

Among the 85 papers included, 20 (24%) were empirical studies (four of which were preprints), and the vast majority (76%) were non-empirical (e.g., perspective papers). Fifty-one papers (61%) had no geographical focus (e.g., were applicable across locations). When they had a geographical focus, most (19 out of 34) addressed the United States (USA) or the United Kingdom (UK) context. While 36 (42%) addressed people with disabilities overall (i.e., had no focus on specific health conditions or impairments), a sizeable amount also has addressed adults with cognitive impairments or intellectual disabilities (*n* = 16), children/youth with disabilities and their families (*n* = 11), and older adults experiencing disabilities (*n* = 9). For the actual content of the disproportionate impacts of lockdown-related measures on people with disabilities, the results of the thematic analysis are provided below.

### Thematic Analysis—Overview

3.1

The thematic analysis of the literature reviewed unravels different types of health and social participation disparities experienced by people with disabilities relative to non-disabled counterparts during the first wave of the COVID-19 pandemic, as a result of lockdown-related measures and regardless of a COVID-19 infection.


Figure 2 provides a graphic display of our themes. The lockdown-related disparities were initially organized in 10 primary themes, with complex inter-links as well as often- blurred limits among one another. Our findings suggest that these 10 disparities are ramifications that stem from two underlying factors, here framed as central or overarching themes.

Some of the primary themes (#1 to #5) address specific disparity types, while some others (#6 to #8) address consequences from the combinations of these disparities. One theme (#9) focuses on the vicious circle of additional disparities (e.g., reduced healthcare access) arising from reduced employment and income. A tenth theme addresses the inaccessibility of tele-solutions for many people with disabilities, which could partly compensate for the other disparities. Finally, all these themes stem from two central themes: the lack of disability-inclusive responses and emergency preparedness, and the structural, socially-entrenched disadvantages people with disabilities were experiencing before pandemic. Each of the themes is covered below.

### Primary Themes

3.2

The 10 primary, lockdown-related disparity themes experienced by people with disabilities during the initial stages of the COVID-19 pandemic are reported below.

#### Disrupted Access to Healthcare (Other Than for COVID-19)

3.2.1

During the first wave of the COVID-19 pandemic, many healthcare services (e.g., outpatient, day services, some in-patient rehabilitation services, assistive devices programs), essential for people with disabilities (e.g., to maintain or recover health and function, manage chronic conditions, prevent secondary conditions, benefit from psychosocial support), were either shutdown or operating at a reduced capacity due to lockdown restrictions. In other words, if they were open, they were operating without sufficient human resources, beds, or other healthcare resources that were diverted to fighting the pandemic [9,14,48,71,80,84,107].

People with disabilities had reduced access to in-person healthcare services due to lack of appropriate transportation during lockdowns [51,73,116]. Access to alternative telehealth solutions was hampered too (details in theme #10), while financial coverage for healthcare services also can be reduced (details in theme #9). Abrupt disruptions, reduced operations, and lack of access to key healthcare services result in risks of deterioration and/or exacerbations of previous conditions and impairments, and may drive a surge in secondary, preventable complications and disabilities [67,84,88,89,92,100-102,112]. One recent survey conducted in Spain with 93 older adults with mild cognitive impairment or dementia, found that 65% of them did not engage in playing memory games during lockdowns [106]. These games are usually conducted in face-to-face memory workshops and mitigate cognitive decline; non-attendance thereby may lead to increased or accelerated cognitive deterioration [106].

People with cerebral palsy faced gaps in accessing healthcare or rehabilitative services which could not be moved to telehealth because they require the utilization of heavy equipment, walking aids, or hydrotherapy services. These services are key to maintaining well-being and to avoiding irreversible contractures and potential deformities [62].

Children with disabilities had diminished access to healthcare services because parents initially feared the consequences of a COVID-19 infection, but also due to the closures of medical settings, caregiving agencies, and also schools which, in many countries often provide therapy services [30,57,62,69,94]. For children with developmental disabilities, lack of timely access to healthcare services can be particularly detrimental. For example, delayed diagnosis and treatment can substantially impact health and development outcomes [86], especially in children with hearing impairments who benefit from early stimulation and interventions [77].

People with severe mental illness may feel threatened by the presence of masked providers, interaction with unknown substitute clinicians, and long wait times among strangers, and therefore may avoid the healthcare system for non-urgent care [55].

People with intellectual or cognitive disabilities hospitalized with any condition during the COVID-19 pandemic may be particularly impacted by changes such as strict visitor policies, as relatives cannot optimally facilitate communication with staff to convey or interpret signs of pain or other symptoms which may be disbelieved or otherwise not noted by staff, especially during busy pandemic times [27,45].

For low- and middle-income countries (LMICs), significant restrictions in healthcare access have been reported. In South Africa, rehabilitation services, assistive device and technology services, therapeutic and developmental interventions, and sign language interpretation services, among others, were not considered essential, and were thereby shutdown during initial lockdowns [73]. In India, disabled migrants or refugees and others without legal documentation faced additional barriers accessing healthcare services under lockdown restrictions [89].

#### Reduced Physical Activity Leading to Health and Functional Decline

3.2.2

People with disabilities, especially older adults experiencing disabilities, may experience sedentary behavior and low physical activity during lockdowns or quarantine periods [84,85,111]. An online, population-based survey in the UK, which involved 5820 adults (4.33% of whom reported having a disability), available through a preprint publication, found that disability from one or more activities of daily living (ADLs) was significantly associated with change toward less intense physical activity behaviors during the initial lockdowns (odds ratio: 2.13; 95% CI 1.87–2.39) [119]. Additionally, an observational study conducted in the UK with participants with chronic pain (*n* = 431) and control participants (*n* = 88), available through a preprint publication, found a greater reduction of physical activity among those with chronic pain (*p* = 0.001) [120].

Any reduction in physical inactivity or person-level deconditioning can lead to novel disability risks and exacerbate existing ones [80,84,101]. Especially in older individuals with neurocognitive disorders or other impairments, a forced reduction of motor/physical activity can cause a progressive loss of personal and instrumental autonomy, as well as a possible worsening of other aging-related clinical concerns, such as sarcopenia, with a subsequent increased risk of falls and other complications [78,84,108].

Finally, people with disabilities have been facing greater barriers in reinitiating sport-related activities post lockdown compared to their non-disabled peers; this may stem from erroneous ableist notions that disability sports may be ‘inferior’, perhaps leading to lowering the priority to reinstate disability specific programs [54].

#### From Physical Distance and Inactivity to Social Isolation and Loneliness

3.2.3

The reduced social participation and loneliness are known risk factors for health-related consequences and have been shown to increase risk of anxiety, depression, malnourishment, dementia and cognitive decline in older adults [63,102,106,112]. Older adults with cognitive and sensory impairments, who are already excluded from social participation due to various reasons, have been asked to distance themselves even further, deepening any existing isolation [76,80,93,106]. For example, those living in long-term care facilities can become separated from one another and the outside world, and thereby experience profound isolation and loneliness [51,63].

In turn, with day activity centres and some sheltered workshops being put on hold during lockdowns, many people with disabilities who live in the community, and rely on these activities, missed out on these opportunities that provided daily structure as well as community participation and social inclusion [78]. People with dementia may be among those who experience isolation due to the disruption of group activities and community support programs [71,84,90].

Apart from older adults with disabilities, people with chronic pain (*n* = 431) who participated in a survey study in the UK reported higher loneliness and tiredness ratings than those in the non-pain group (*n* = 88), and reported being more likely to self-isolate to protect themselves from contracting the virus (both *p*’s < 0.001) [120].

#### Disruption of Personal Assistance and Community Support Networks

3.2.4

People with disabilities who rely on caregiving or personal assistance to meet their needs experienced interruptions or discontinuation of this assistance when they were required to quarantine [27,28,59] or when their caregivers/ personal assistants were quarantined or fell ill, and were unable to provide continuous support [27,65].

Moreover, people with disabilities who live in the community but need personal assistance experienced difficulties accessing required medication, food, or assistance in ADLs (e.g., bathing, dressing, feeding, and toileting needs) [49,72,80,81,84,85,95,101,116].

A preprint survey study in the UK, conducted during the lockdown in April and involving 2,597 participants aged 70 and over, found that amongst the 511 respondents who had reported difficulty in performing at least one ADL prior to the pandemic, just seven respondents reported receiving help with basic personal needs like dressing, eating, or bathing during the first four weeks of the lockdown. Also, the only subgroup which was likely to receive less help (i.e., 10% less during the lockdowns) were the older people in the sample who also reported difficulty with two or more instrumental ADLs (e.g., going shopping, doing laundry, paying bills) prior to the pandemic [118].

Without assistance, including transportation, many people with disabilities living in the community had difficulties leaving their homes to go outside to purchase basic goods (e.g., food, meals) [51,73,75]. In turn, delivery of basic goods to their home, which many people with disabilities usually rely on in order to remain independent, became less reliable by the sudden increased demand under lockdown conditions [71]. Communicating with people who use opaque masks, preventing lip reading, became harder for people with hearing impairments [75]. People with visual impairments often use either close assistance or touch to navigate public space, including braille signs and shared surfaces such as handrails, door handles and handrails on public transport. For these individuals, compliance with infection prevention measures often implied not leaving their homes [64].

People with dementia stopped taking their medications, either because they run out and were unable to refill them, or because of a lack of assistance from formal or informal caregivers they rely on for compliance with medication regimens [71]. Without assistance, some people with dementia have greater risk for falls or aspiration pneumonia [112].

For people with disabilities the disruption of social and community support networks often resulted in hospitalizations without medical necessity, sometimes referred to as ‘social hospitalization’, or short-term stays in residential facilities, which, in turn, aggravated the risks of COVID-19 infection [72]. Similarly, people with developmental disabilities remained in local hospitals past the point of medical necessity because of the lack of sufficient support conditions to return them home safely [70].

#### Children with Disabilities Are Disproportionally Affected by School Closures

3.2.5

Children with complex physical needs and impairments depend on access to educational equipment and professional support, and, for many children, this support is only available through school [117]. With school service providers and support systems unavailable, students were sent home with learning packets, which many children with disability and their parents found it hard to complete without professional support [51]. Additionally, with limited parental monitoring (e.g., parents teleworking at the same time), in unstructured environments, youth with disabilities have increased likelihood of engaging in sedentary activities, such as increased screen time [30].

Disruption of therapies and school services for children with developmental disabilities have led to greater stress and regression in skills compared to non-disabled counterparts [18,30,86]. Students with disabilities usually benefit from tailored, structured, and often multi-sensorial educational strategies, and therefore are especially vulnerable to regression when these services are removed, reduced, or modified to be delivered by telematic forms [30,109]. Some children with disabilities who rely heavily on structure and daily routines (e.g., children with autism spectrum disorder) struggled with the disturbed routines due to the absence of school and therapy services during lockdowns [15,49,92,94]. Hence, they often responded with increased severity or intensity of challenging behaviours [61].

As school services often provide a reliable source of meals, learning opportunities, social participation, and may serve as supportive environments for the most vulnerable children with disabilities (e.g., young refugees, young migrants, children living without parental care, homeless children, children living in urban slums, children in conflict-affected areas), the closures of schools and inherent support systems (see theme #4) risks bringing the greatest drawbacks and aggravate existing social and educational disparities for many children with disabilities [30,69].

#### Psychological Consequences Arise from Disrupted Routines, Activities, and Support

3.2.6

People with disabilities can be especially vulnerable to numerous psychological consequences of disrupted routines, activities, and support networks (see aforementioned), as well as stress and anxiety from the fear of contracting an especially harmful COVID-19 infection or from a general lack of understanding of the pandemic and its restrictions.

For instance, under isolation rules, or with prolonged periods indoors, people with dementia may be without their usual access to community support programs and familiar routines. This can cause them to become anxious, angry, stressed, and agitated [71,84,90], and lead to an increased risk for suicidal behaviors [112]. Facing an unknown situation, a common source of stress in the context of the pandemic, may be even more impactful for individuals with cognitive impairments [71,106]. For people with dementia, sleep may be further disrupted due to anxiety and loss of social rhythms, which is compounded by a lack of activities and stimulation. These disruptions may exacerbate delirium and accelerate cognitive decline [71]. Furthermore, people with cognitive impairments can become confused and disorientated by interactions with caregivers or healthcare providers wearing masks and protective eyewear [47].

For adults or children with intellectual disabilities, it can be hard to understand the necessity for the restrictions, e.g., why they can’t receive a hug from a caregiver, which’ may lead to an increased anxiety, agitation, and challenging behavior [51,78,86,94,99]. In a survey study with additional qualitative analysis in the UK, parents (*n* = 241) described situations in which a low level of understanding of the pandemic by their children with disabilities led to distress because they could not understand why everything had changed. In the cases of minimally verbal children, disorientation was sometimes expressed through challenging behavior [31]. When caregivers of children with disabilities need to self-isolate, the change of carers can also generate stress and exacerbation of behavioural problems [46,78,99].

Individuals with cerebral palsy often have higher rates of anxiety and depression, which may worsen during a pandemic due to lack of access to regular therapy schedules, while both increased stress and lack of access to in-person therapy may worsen their hypertonia [62].

Individuals with autism spectrum disorders may experience emotional problems, acute anxiety, and disrupted behavior as a result of the disruption of carefully established routines [46,51,117]. People with autism spectrum disorder who can understand information about COVID-19 can become over-focused and subsequently overwhelmed by the amount of information, risks, and preventive measures, which may heighten their levels of anxiety and paranoid thinking [46].

For people with severe mental illness, there is heightened risk of relapse because of high susceptibility to stress under lockdown measures and an overall reduced ability to cope with stress in disaster situations compared to the general population. For instance, from a sample of 132 persons with severe mental illness surveyed in South India, around 30% of those who were stable before lockdown had a relapse and 22% stopped their psychiatric medication due to lack of access to medication and mental health professionals [104]. Often, the relapse translated into poor hygiene, inability to practice social distancing, delay in seeking medical attention, aggression and increased substance use, as well as suicidal behavior, the latter expressed by 14.4% of those surveyed [104].

An observational, study (preprint) conducted in the UK found that people with chronic pain (*n* = 431) self-reported increases in anxiety, depressed mood, and pain catastrophizing compared to a sample of control participants (*n* = 88; all *p*’s < 0.01) [120].

A clinical interview and survey study in Southern Italy with persons with amyotrophic lateral sclerosis found that one out of five patients in a sample of 32 experienced a significant worsening of quality of life since the start of the quarantine due to behavioral and sleep disturbances [110]. Behavioral disturbances (anger attacks) and sleep disturbances (difficulty falling asleep, frequent awakenings) were reported in 15% and 20% of respondents, respectively [110].

In a cross-sectional survey of 269 web-literate persons with self-reported disabilities and chronic conditions in the United States, moderate levels of stress, depression, and anxiety were found on average, while coping strategies explained a total of 54% of variance in well-being [105]. After controlling for demographic and psychological characteristics, participants who had high ratings on active coping, use of emotional support, humor, religion, and low ratings on self-blame were found to have high ratings on well-being [105].

A study in the Netherlands about the utilization of an online support service found that people with intellectual disabilities living independently were contacting the online support service more often, especially during the first weeks of the pandemic, because they were considerably worried and experienced high levels of anxiety [113].

Many people with disabilities are worried and fearful about the possibility of being vulnerable to COVID-19 and its consequences, and that they may not receive equitable healthcare because of their disability [52,79,116]. This fear was intensified by early discussions of the need to ration life-saving medical equipment [45,52]. Hence, the uncertainty about access to life-saving treatments and their awareness of existing bias and disability stigma (including inaccurate ableist assumptions about their quality of life, with impact on healthcare decisions related to medical rationing) can create anxiety, distrust, and overall psychological harm to many people with disabilities [18,57,66,67].

#### Family and Informal Caregiver Burden and Stress

3.2.7

Family and caregiver burden increased as usual supports of residential schools, day services, respite care or overall community support services for people with disabilities living partly or full time in the community were unavailable [46,78,90,106]. An increased number of families needing caregiver support for anxiety and uncertainty was observed as day programs were closed and stay-at-home orders enforced [68].

As many residential institutions closed, residents were obliged to return to their families, many of whom lacked the time or means to provide proper care [51]. For those facilities that remained open, disruption in many support and day services for people with disabilities (see theme 4) resulted into prolonged hours of caregiving in the context of decreased psychosocial support for informal caregivers [84]. In addition, families were often not permitted to visit or even communicate with busy staff in either residential or hospital facilities, leaving many families without any information on the status of their family members with disabilities [68]. Many families also worry that the lives of their family members with disabilities may be devalued and that they may face disadvantage in any rationing decisions due to disability stigma [52].

School closures added further stress to parents already worried about the pandemic [94]. As rehabilitation and school services shut down, parents were experiencing insecurity, abandonment, and anxiety, as they often did not feel equipped to provide their children’s special education needs. Parents struggled to provide the same level of academic support without relevant training and expertise [57], and struggled with disruptive behaviours from their children as a result of disrupted routines [109]. In turn, children with certain developmental disabilities like autism spectrum disorder may lack the cognitive flexibility to understand that parents were trying to play the role of their teachers or therapists during some parts of the day, which added complexity and stress to these tasks and everyday life [57].

A qualitative study with 241 parents or carers of school-aged children in the UK generated several accounts of single parents who were isolated during lockdowns with a child with disabilities who displayed disruptive behaviour without access to any of the support and respite that usually would help them to fulfill their parental role effectively. These feelings were exacerbated by worries about who would look after the child if the parents died as a result of COVID-19 [31].

#### Risks of Maltreatment, Violence, and Self-Harm

3.2.8

The COVID-19 pandemic magnified existing barriers facing people with disabilities who are experiencing interpersonal violence [74]. These barriers include reliance on the perpetrator for care and assistance, difficulty reporting abuse and seeking help, and fear of retaliation and other negative consequences if abuse is reported [74].

By isolating older adults with disabilities from community support networks, they remain in closer contact with their caregivers, under stressful circumstances, which increases susceptibility to violence, abuse and neglect [89]. With care and support being restricted by the pandemic, people with dementia, for example, have higher chances of being subject to neglect and abuse [84] and to develop delirium and aggressive behaviors leading to self-injuries [112].

Children with disabilities are also at a great risk for maltreatment due to the closure of schools or child protective services (e.g., which exert some social control of these risks), disrupted routines, socio-economic strain within the family environments, and/or an limited ability to communicate [30,86]. Also, restriction of travels between households, with consequent loss of support from extended family members, adds to the challenge of parenting children with disabilities in lockdown contexts [57]. The disruption of routines among children with disabilities can lead to self-harm [61].

Empirical research from a pool of 44,775 participants in the UK surveyed during March—the first month of the COVID-19 pandemic—indicated that 7.0% reported a disability. In turn, compared to the whole sample, those with a disability reported higher levels of: psychological abuse (18.4% versus 8.3%), physical abuse (9.2% versus 2.9%), self-harm/suicidal thoughts (48.0% versus 17.8%), and self-harm behaviors (17.8% versus 4.9%) [103].

In South India, a telephone survey conducted during the initial phase of the pandemic found that, among a sample of 132 people with severe mental illness, 63.6% reported they were experiencing verbal and physical aggression from others [104]. In an April 2020 virtual meeting of physicians specializing in pediatric rehabilitation medicine, anecdotal reports existed of increased referrals for non-accidental trauma affecting children, suggesting a risk of increasing domestic violence and abuse possibly arising from contextual variables such as families under financial stress from employment challenges, psychosocial stress from being isolated from the community support structure, and overall anxiety about the ongoing effects of the COVID-19 pandemic [58].

In the context of telehealth support, it may be difficult for people with disabilities to truly be in a private location when talking to providers as a means to report abuse [74]. Given that people with disabilities already experience lack of employment much more often than those without a disability, the financial consequences of this crisis (see the following theme) may be magnified and lead to both increased reliance on the perpetrator and increased difficulty in mitigating the effects of abuse because of a lack of financial resources [74].

#### Reduced Employment and/or Income Exacerbating Disparities

3.2.9

Many people with disabilities have lost their jobs because of the pandemic, which can put them at economic hardship. Since small businesses and non-profit organizations are mostly closed, their employees with disabilities, some under supported employment, have been furloughed indefinitely, while many workers with disabilities have not been transitioned for remote work [51,80]. People with disabilities in LMICs, especially woman, often work in the informal sector, facing food insecurity and the absence of sick leave or unemployment benefits [14]. Economic implications of the pandemic lead to loss of employment predominantly for those with precarious jobs, in which people with severe mental illness or other disabilities are overrepresented, thus adding financial stresses, housing and food insecurities [55].

In Australia, the Coronavirus Supplement paid to those receiving unemployment benefits have excluded those receiving Disability Support Pension [64]. In Chile, disabled people who received a disability pension were not entitled to the COVID-19 cash transfer that was meant to help the most vulnerable populations [115], even though expenses likely increased as a result of the pandemic [14,115]. For instance, as telehealth platforms become commoditized, the accommodation costs may be transferred to vulnerable populations such as people with disabilities, who may not be able to afford them [29]. Moreover, people with disabilities in South America might not be able to navigate the typically complex documentation processes for obtaining compensation for these accommodations [115]. This is an issue that has been common across many LMICs [14]. Overall, the lack of welfare protection for many People with disabilities in South America, previously in place, have been exacerbated during emergency situations [115].

In many LMICs, a large proportion of people with disabilities live in single-income households (e.g., household members may forgo work to provide caregiving support), therefore the COVID-related unemployment (e.g., of the person earning the single household income) provide economic hardship [14]. Furthermore, while the allotment of people with disabilities-targeted cash transfers is often controlled fully by others in their household, people with disabilities’ entitlements may not be used for their own sake [14]. Finally, people with disabilities who became unemployed during the pandemic may also take longer to re-enter the workforce with the ease of restrictions, due to stigma, inaccessible environments, and poor access to education and training that limit job opportunities [14].

In some countries, the loss of medical health coverage or benefits associated with employment is also of concern for people with disabilities, as this reduces one’s ability to pay for prescription medication, hence causing non-adherence or even discontinuation of key medication regimens [55]. In addition, there are extraordinary pressures on public budgets from increased spending and reduced tax revenue as a result of the economic consequences of the pandemic. This may put many people with disabilities, especially those of lowest income, at risk of not being able to access safety-net services such as those under Medicaid in the US due to stricter eligibility criteria [83,101]. Furthermore, operations of Medicaid-funded nursing homes and home care services are likely to be greatly affected because low-income and African-American communities disproportionately represent the direct care workforce and these communities have experienced higher rates of infection [101]. The increasing unemployment and loss of health insurance resulting from the pandemic threatens healthcare access for children with disabilities [30].

#### Digital Divide in Access to Health, Education, and Support Services

3.2.10

People with disabilities often have had greater difficulties accessing or benefiting from these services due to lack of access to or accommodations in digital solutions. During the pandemic, the consequences of this lack of access has been exacerbated since telematic forms of service delivery and support have become more widely used as a complement or replacement of in-person services and care.

Many people with dementia had no access to the Internet or an electronic device, or sometimes no supporter (e.g., a family member) to assist them with the use of the technology to access remotely-delivered support services or care (e.g., cognitive stimulation) [90]. Similarly, access to telehealth services for people with cerebral palsy during the pandemic often involved the need for mediators for these services to be accessible [62]. Exclusion criteria for telemedicine visits often include the inability to provide informed consent, which can prevent the use of telemedicine for people with advanced cognitive dysfunction, and those who need an interpreter, for example [87]. In turn, cognitive and sensory impairments reduce the ability to provide seamless care via video visits if no proper accommodations are provided [56].

Older adults experiencing disability are among those who need special protection during the pandemic, such as physical distancing, and hence may benefit from telematic services. However, many older adults experiencing disability face challenges with the access to and the usability of mobile information and communication technology [96]. Broadband fast internet is inaccessible in many rural and low-income communities as well as in developing countries, thereby telemedicine access for people with disabilities living in these communities is suboptimal [29,89]. Children with disabilities living in poverty might not have electronic equipment or access to tele-schooling activities [30]. People with severe and persistent mental illness may not have access to the internet or the literacy skills to benefit from telehealth solutions [55].

In Southern Italy, video visits were offered as a telehealth solution, but refused by many people with amyotrophic lateral sclerosis because the large majority of participants did not own a computer or smartphone but rather only a cell phone; this digital divide limited physical examinations via telehealth [110]. Apart from the pandemic, the swift transition of primary healthcare provision from in-person to tele-consultations has led many people unable to access services for regular check-ups, presumably due to lack of literacy and access to appropriate technology [98].

The lack of universal design and web accessibility standards in telemedicine platforms often exclude people with disabilities [29,114]. Overall, there is a lack of accessibility extensions such as screen readers, sign language, captions, magnification, color, and contrast [29]. Similarly, most telemedicine platforms do not have custom features to facilitate healthcare communications for persons who are deaf or blind or for persons with cognitive impairments, and there is a dearth of health education materials for persons with language and literacy challenges [29]. Finally, providers who are utilizing telemedicine may not understand and be able to address the accessibility issues even if the systems are designed correctly [29].

The digital divide does not apply only to missed opportunities for telehealth solutions. During lockdowns, many people with disabilities could not make online purchases as they may not have credit cards, Internet, or electronic devices, or due to the lack of universal design of relevant websites [73]. College students with special needs faced added difficulties in terms of accommodation and online virtual learning (e.g., students who need paper and pencil tests or assistive technologies to access testing materials cannot necessarily test in their usual ways) [60].

### Central Themes–Underlying Contributors

3.3

#### Lack of Disability-Inclusive Response and Emergency Preparedness

3.3.1

The presence of disability-inclusive emergency pandemic preparedness could have prevented or mediated aforementioned disparities, at least partly. Yet, the reviewed literature emphasized either a limited or no emergency or contingency planning addressing people with disabilities’ needs, applicable to the pandemic situation [49,69,91,116].

One of the reasons for lack of preparedness is lack of data (e.g., surveillance data) on people with disabilities, which has been limited both before and during the pandemic [45,49,69,97]. Although disability status should be considered important demographic information (e.g., to assess any disproportional impacts), these data are not systematically collected or included in official reports [50]. Current COVID-19 estimates among people with disabilities have come from assisted living facilities, in which the disability status of residents is sometimes documented [45,97]. However, these data represent only a fraction of the population with disabilities [45]. Failure to accurately record disability status on the death certificate prevents the understanding of the full effect of the pandemic on this population [50]. As a result, there is little information that allows public health experts to assess the impact of the COVID-19 pandemic on people with disabilities and the appropriate, equity-oriented public health and policy responses [45].

Despite recommendations for local governments to include people with disabilities in the planning, integration, and implementation of emergency programs (e.g., regarding the access to education, employment, and healthcare services), people with disabilities haven’t been typically included [49,82]. In South Africa, although COVID-19 disaster management committees were established prior to issuing lockdown measures, no disability advocates were involved, which may have contributed to the negligence of disability-related issues in COVID-19 responses [73].

#### Structural, Pre-Pandemic Disparities Exacerbated during the Pandemic

3.3.2

As a seminal theme on the intricate net of causes of the aforementioned disparities (Figure 1), it is important to recognize that people with disabilities faced structural, socially-determined disadvantages pre-pandemic, and that these have been exacerbated during the pandemic.

It has been noted that the lack of timely access to quality healthcare is a structural disparity commonly faced by people with disabilities, aggravated and exposed during the pandemic [95]. Similarly, it has been argued that wider disparities faced by people with disabilities during the COVID-19 pandemic arise from pre-pandemic discrimination, marginalization, ableism, ageism, sexism, and stigma leading to human rights and social participation deprivation among many people with disabilities [89]. The exclusion of people with disabilities and disability-related issues among disaster or emergency preparedness is a long-standing issue [49]. The limited surveillance data and data on People with disabilities’ needs, unmet needs, experienced disparities or circumstances has been limited both before and during the pandemic [45,49,69,97].

By the same token, the lower priority attributed to disability sports pre-pandemically was reflected into the way sports were considered in a pandemic context, and possible in the future, as well [54]. The COVID-19 pandemic magnified the interpersonal violence People with disabilities often experience [74]. The divide in the access of digital solutions, with either universal design or specific accommodations, has been an issue for many people with disabilities prior to the pandemic, albeit possibly more devastating as the society needed to further rely on tele-health, tele-work, or tele-schooling solutions [29,80]. Reduced employment and income is common among people with disabilities in non-pandemic times, and might have been aggravated during the pandemic [80,115]. There is a need to examine the underpinnings of existing health disparities and the values and beliefs of existing social and political systems that created inequities for people with disabilities [79]. These structural disadvantages are being further experienced now, but likely will remain after the pandemic if the opportunity for reform is not seized [80,82].

Overall, this review suggests that the pandemic has exacerbated disparities faced by people with disabilities. The ramifications of these disparities essentially reflect structural societal barriers that require transformational change in societies, not merely responses that mitigate the exacerbation of disparities during major public health crises.

## Discussion

4

This scoping review synthesized a whole range of inter-linked health, social participation, and socio-economic disparities that people with disabilities experienced during the first wave of the COVID-19 pandemic, as a result of lockdown-related measures. People with disabilities experienced restricted access to health, education, and community services that are essential for them, including meeting basic life and functional needs. They also experienced risks of maltreatment, psychological consequences (e.g., from disrupted routines and activities), and difficulties accessing digital solutions that are not inclusive of their needs. Their families and informal caregivers also experienced a disproportionate burden and stress. In turn, reduced employment and income exacerbated existing socio-economic disparities, and limited access to needed services. This analysis suggests that all these pandemic disparities arise from the lack of disability-inclusive responses and preparedness, and seminally from socially-determined disparities that people with disabilities have been experiencing for long time.

Lockdown-related disparities faced by people with disabilities showed to be manifold, significant, and intricate. As a result, explicit public health and policy responses aimed to prevent or mitigate them are necessary, and would need to address health, social participation, and socio-economic disparities in tandem. When problems are systemic, the solutions must be too. An integrative development of health and social policies is needed in the current pandemic context [121,122], and these integrated policies should be disability-inclusive [14,80,81,95,115]. Disability-inclusive plans to prepare for and respond to a pandemic seems absolutely required, informed by research on typical needs and disparities faced by people with disabilities, such as those here synthesized, and involving people with disabilities and their advocates in their development and monitoring [22,27,65,69,73,75,89].

The distinct themes and sub-themes identified and presented through this analysis are intricate and interconnected. For example, added risks to maltreatment, negligence, or abuse toward people with disabilities arise due to service closures, across a range of the health, educational and social sectors. These closures removed key supportive services and part of the societal control over maltreatment, which could not be replaced by digital solutions that are not designed to be used independently by people with disabilities. In turn, the disruption of extended family or community support networks also contributed to an increased caregiving burden, which can turn maltreatment more likely. Finally, harder economic conditions can also lead to increased family stress and a greater reliance of people with disabilities in the perpetrator, within the household. These factors, addressed in multiple themes, illustrate how themes can be inter-dependent.

Indeed, in this context of multiple, mutual, and intricate relationships, effective policy responses cannot be fragmented, e.g., need to be intersectoral, to address, at the same time, the whole set of factors that contribute to disability disparities.

Themes addressed here were not specific to impairment type. For example, even when children with disabilities were addressed by a specific theme, related to schooling, this involved the lifespan and a related occupation rather than impairment or disease categories. A similar rationale applied to the context of social isolation, lack of (physical) activity, or loneliness of older adults with disabilities. When specific impairments or diagnostic categories were addressed within a theme, regarding a specific vulnerability, the theme applied similarly to other subgroups of people with disabilities facing comparable social circumstances regardless of impairment type. This underscores the importance of structural and social determinants, including social determinants of health, rather than the individual, impairment, or disease nature of the identified disparities.

From the central themes in our analysis, it is suggested that lockdown-related disparities experienced by people with disabilities often arise from structural disparities that people with disabilities were experiencing before the pandemic, and were exacerbated thereafter. In turn, the structural disparities often arise from stigma, ableism, discrimination, and marginalization of people with disabilities-still prevalent in societies, which lead to the social exclusion of people with disabilities and have contributed to the diverse disability disparities observed during the COVID-19 pandemic [15,79,89,123]. Hence, fundamentally addressing the disability stigma and discrimination in societies can contribute to address the structural determinants of disparities experienced by people with disabilities, during pandemics and beyond.

Many health and support services for people with disabilities might be considered essential during lockdowns, and contingency plans should be in place (e.g., in institutions, municipalities, official agencies, civil society) for mitigating any disruption in community or professional support for people with disabilities. These and other responses can address the pandemic-specific ramifications of disability disparities. However, these actions should be supplemented by more fundamental societal changes for a systemic, extra-pandemic use of universal design principles (e.g., not only in architecture or urban planning but also in others sectors such as health policies and social policies [124]) as well as by any specific accommodations (e.g., accessibility options [29,114]). The pandemic challenges have turned existing disability disparities more noticeable, hence opening an opportunity for systematic action that must be seized. For example, the development of disability-inclusive telehealth, telework, and other digital platforms may increase the livelihood, health, participation, and social inclusion of people with disabilities for the ‘new normal’ after the pandemic [29,62,80].

Our results describe lockdown-related disparities often experienced by community-dwelling people with disabilities. In our first scoping review based on the same data, we found a unique vulnerability (for health risks and consequences of a COVID-19 infection) amongst people with disabilities living in residential or long-term care settings [8]. Greater levels of social isolation of people with disabilities living in the community can partly protect from infection risks, but at the cost of other disproportionate health, social, participation, and socio-economic unmet needs. Pandemic control policies need to account for both infection and lockdown-related risks, and include effective counter-measures that prevent or mitigate the unintended, disparate, and systemic consequences for people with disabilities of any needed lockdown measures [4,18].

Finally, only a few of the included papers addressed specifically the LMICs. One of the likely reasons is the paucity of disaggregated data for disability in many LMICs, which prevents the accurate identification of the many type of disparities likely experienced by people with disabilities in LMICs during the COVID-19 pandemic. Strengthening data systems, across sectors, to ensure disability-specific, readily available data could be the first step toward disability-inclusiveness in public health and policy responses to emergency events, and beyond to address established disparities people with disabilities typically face in LMICs in non-pandemic times, as well.

### Limitations

These results should be interpreted in the light of the following limitations: This review addressed only the peer-reviewed or preprint literature (excluding the grey literature) available up to mid-September 2020, roughly equating to data and perspectives from the first-wave of the COVID-19 pandemic. Also, only a few empirical studies have been found, longitudinal data is lacking, and only a small fraction of the included papers address the LMICs. It is important to develop further studies and systematic or scoping reviews, with extended time and geographical coverage, in these important but seemingly under-researched matters.

As typical in scoping reviews, this work did not involve quality appraisals of methods, which combined with the presence of preprint studies, albeit signposted, lead to careful interpretation of the few existing studies. Non-empirical papers were included, which provided key rationales and occasional qualitative accounts, or illustrative examples, of disparities faced by people with disabilities. These were essential elements to build our thematic results in addition to, and in dynamic complement with, the empirical literature; however, these perspectives should be carefully interpreted. The same applies to the preprint studies, which were not peer-reviewed; therefore, their findings should be interpreted with caution, as well.

Finally, although we discuss some strategies are provided to policy-makers and public health stakeholders (e.g., considering the multiple and inter-linked disparities at the same time for an integrative planning) to prevent or mitigate the identified disparities, here we do not address a set of actions that can be taken for disability-inclusive preparedness and responses to pandemic events.

## Conclusions

5

Lockdown-related measures to contain the COVID-19 pandemic can disproportionally affect people with disabilities in health, educational, social support, social participation, and socio-economic terms. These disparities influenced one another, and arguably so as public health inequities and occupational injustices are often determined by social determinants of health and occupation [34,35,121]. Hence, public health and policy interventions, including social policies, might be planned and coordinated across sectors, and address the whole range of mutually-reinforced, lockdown-related disparities that people with disabilities have been experiencing during the COVID-19 pandemic and beyond. Indeed, our review of lockdown-related disparities also determined that lack of disability-inclusive response and emergency preparedness as well as that pre-pandemic disparities created structural disadvantages which were further exacerbated during the pandemic. Both structural disparities and their pandemic ramifications need to be addressed by disability-inclusive public health and policy measures.

## Supplementary Material

The following are available online at https://www.mdpi.com/article/10.3390/ijerph18126178/s1, Supplementary 1: Working definitions and search strategies. Supplementary 2: Data extractions and the reviewers’ brief synthesis of each paper.

Supplementary files

## Figures and Tables

**Figure 1 F1:**
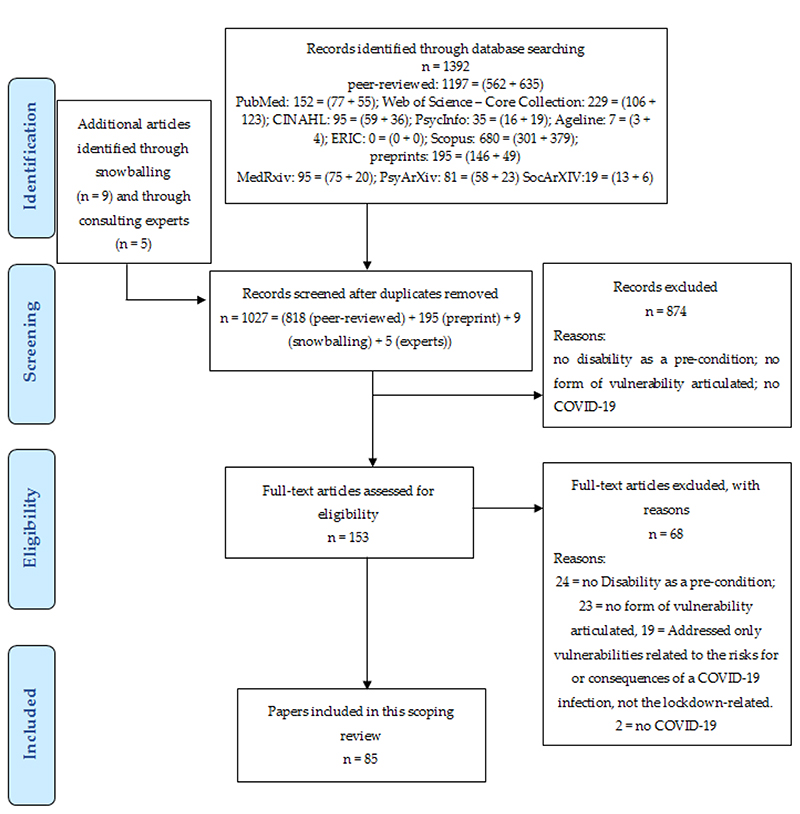
PRISMA flowchart of the scoping review with thematic analysis.

**Figure 2 F2:**
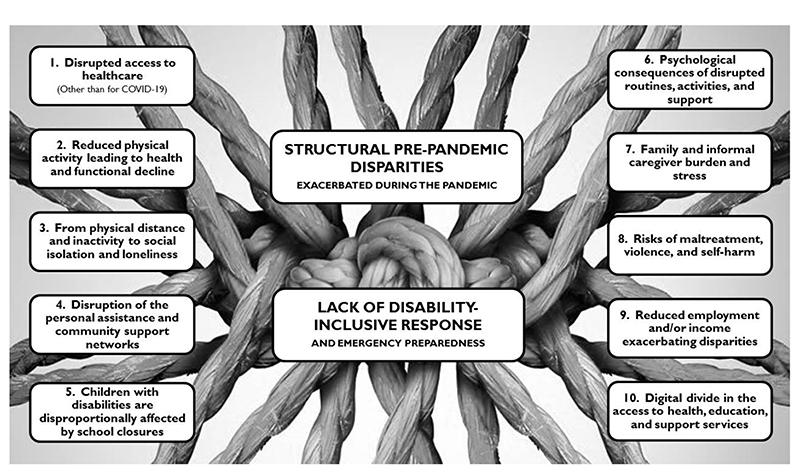
Graphic display of the themes.

**Table 1 T1:** Quantitative map of the literature analyzed.

Characteristics	Number (%)	Citations
PUBLICATIONS TYPE AND SOURCE		
Perspective papers (e.g., viewpoints, commentaries, essays, ethics/advocacy)	43 (51%)	[9,14,15,27,29,30,45-81]
Narrative summary/review (non-systematic)	8 (9%)	[82-89]
Editorial or Letter to the editor	8 (9%)	[90-97]
Framework or Experts recommendations	6 (7%)	[18,98-102]
Non-empirical (peer-reviewed): SUB-TOTAL	65 (76%)	
Cross-sectional surveys	4 (4.5%)	[103-106]
Institutional case report	4 (4.5%)	[28,107-109]
Pilot feasibility study	2 (2.5%)	[110,111]
Ecological study	1 (1.2%)	[112]
Survey research, with qualitative analysis	1 (1.2%)	[31]
Quantitative analysis of contacts to support services	1 (1.2%)	[113]
Analysis of webpages on accessibility compliance	1 (1.2%)	[114]
Documentary research and framework analysis	1 (1.2%)	[115]
Country case report	1(1.2%)	[116]
Empirical studies (peer-reviewed): SUB-TOTAL	16 (19%)	
Survey research	3 (3.8%)	[117-119]
Comparative cross-sectional survey (control group)	1 (1.2%)	[120]
Preprint studies: SUB-TOTAL	4 (5%)	
GEOGRAPHICAL FOCUS		
No geographical focus (e.g., applicable across locations)	52 (61%)	[15,18,27,29,45,46,48-52,54-57,59-67,69-72,74,76-80,85,86,88-100,102,112,114]
United States of America (USA)	10 (12%)	[28,30,58,68,83,87,101,105,109,111]
United Kingdom (UK)	9 (10%)	[31,47,75,81,103,117-120]
Low- and Middle-Income countries (LMICs)	3 (3.8%)	[9,14,53]
Spain	2 (2.5%)	[106,107]
Italy	1 (1.2%)	[110]
Netherlands	1 (1.2%)	[113]
Singapore	1 (1.2%)	[108]
South Korea	1 (1.2%)	[116]
South Africa	1 (1.2%)	[73]
Philippines	1 (1.2%)	[82]
India	1 (1.2%)	[104]
Asia	1 (1.2%)	[84]
Latin America	1 (1.2%)	[115]
HEALTH CONDITIONS		
People with disabilities, Overall	36 (42%)	[9,14,15,18,27,29,45,46,48-52,54,64-66,69,70,73-75,79-83,92,95,97,98,103,105,114-116,119]
Adults with cognitive impairment (e.g., dementia) or intellectual disabilities	16 (19%)	[28,47,56,59,68,71,76,78,90,93,99,106,107,112,113]
Children/youth with disabilities (and their families)	11 (13%)	[30,31,53,57,58,77,86,91,94,109,117]
Older adults experiencing disabilities	9 (10%)	[63,84,85,89,96,101,102,111,118]
Severe Mental Illness	2 (2.5%)	[55,104]
Spinal Cord Injury	1 (1.2%)	[72]
People with disabilities living in residential or long-term facilities	1 (1.2%)	[108]
Visual impairments	1 (1.2%)	[67]
Autism Spectrum Disorder	1 (1.2%)	[67]
Cerebral Palsy	1 (1.2%)	[62]
Cerebellar Ataxia	1 (1.2%)	[100]
Amyotrophic Lateral Sclerosis	1 (1.2%)	[110]
Parkinson’s	1 (1.2%)	[88]
People recovering from joint surgery	1 (1.2%)	[87]
Chronic pain	1 (1.2%)	[120]
College students with special needs	1 (1.2%)	[60]

## Data Availability

Not applicable.

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
