# Peer review of "Lockdown-Related Disparities Experienced by People with Disabilities during the First Wave of the COVID-19 Pandemic: Scoping Review with Thematic Analysis"

_ijerph, 2021, doi:10.3390/ijerph18126178_

Round 1
Reviewer 1 Report
The authors present an interesting scoping review highlighting very important points related to health and social impacts on people with disabilities because of lockdown strategies. The discussion opens the debate about what is “essential service” and how we could organize, as a society, different ways to provide “essential services” for different groups, such as people with disabilities. To advance in this discussion, one important point in my opinion is defining what is ”health and social participation disparities”. The authors propose to analyse the literature based on this, however there is not a definition about health and social participation in the article. And I think it would be contributive to add a concept about this to continue the discussion beyond this article: how we can define essential services and how to create alternatives to include different strategies to approach many difficult consequences from lockdown.
Another important point is the assertion: “the lack of disability-inclusive responses and emergency preparedness, and the structural, socially entrenched disadvantages PwD were experiencing before pandemic”. This is very important and pushes the discussion about social services and the necessity of social benefits for many people in different countries, including people with disability in social vulnerability situations. I would suggest including in the Conclusion the social policies as well (not just public health) because it is strongly connected with the structural disadvantages experienced by different groups, such as people with disability.
Finally, I would suggest reviewing the references list, for example 8 and 15 are the same.
Congratulations for this article and I hope it can bring more discussion about difficult life in pandemic times.
Author Response
Response to the Reviewers
We would like to extend our gratitude to all the reviewers for the rapid and precise feedback, which helped us to improve the quality of the manuscript.
We respond below to each comment and highlight any resultant change in the manuscript. We do so in blue, while the reviewers’ comments are transcribed in black.
Reviewer 1
The authors present an interesting scoping review highlighting very important points related to health and social impacts on people with disabilities because of lockdown strategies. The discussion opens the debate about what is “essential service” and how we could organize, as a society, different ways to provide “essential services” for different groups, such as people with disabilities.
Thank you.
To advance in this discussion, one important point in my opinion is defining what is ”health and social participation disparities”. The authors propose to analyse the literature based on this, however there is not a definition about health and social participation in the article. And I think it would be contributive to add a concept about this to continue the discussion beyond this article: how we can define essential services and how to create alternatives to include different strategies to approach many difficult consequences from lockdown.
Early at the Introduction, we further highlight what we mean by different types of health disparities (e.g., health status, health outcomes, and healthcare access) that PwD have been experiencing for long time, yet have been exacerbated during the pandemic (new content underlined).
“PwD may be disproportionally affected by the COVID-19 pandemic. This disproportional impact entails greater risks of being infected (especially for PwD living in residential or long-term care institutions) or greater risks of having severe health consequences once infected, including from unethical disadvantages in the access to life-saving treatments.[4] Furthermore, PwD often require routine health and rehabilitative care (e.g. at home, outpatient) to maintain or recover their health and function. However, many of these services were considered non-essential, closed or functioning with important human and physical resources restrictions during initial lockdowns to contain the pandemic.[5,6] There are widely documented health status, health outcomes, and healthcare access disparities that PwD have been experiencing for long time[7-9]. Yet, these forms health disparities have been widened and exacerbated during the COVID-19 pandemic. [4,10,11]”
Later in the Introduction, after some examples of social participation disparities, we now also highlight the following (all content is new):
“Overall, PwD regularly experience occupational injustices and social participation disparities when they are denied, excluded, or deprived from an equal opportunity to pursue meaningful occupations, social roles, and social integration, when compared to people without disabilities.[34,35] In turn, these social participation disparities directly contribute to broader health disparities, as participating in meaningful occupations is a key determinant of human health and wellbeing.[36,37]”
Another important point is the assertion: “the lack of disability-inclusive responses and emergency preparedness, and the structural, socially entrenched disadvantages PwD were experiencing before pandemic”. This is very important and pushes the discussion about social services and the necessity of social benefits for many people in different countries, including people with disability in social vulnerability situations. I would suggest including in the Conclusion the social policies as well (not just public health) because it is strongly connected with the structural disadvantages experienced by different groups, such as people with disabilities.
We appreciate the suggestion and our conclusion now begins to read like it follows (new content underlined).
“Lockdown-related measures to contain the COVID-19 pandemic can disproportionally affect PwD in health, educational, social support, social participation, and socio-economic terms. These disparities influenced one another, and arguably so as public health inequities and occupational injustices are often determined by social determinants of health and occupation [36,37,125]. Hence, public health and policy interventions, including social policies, might be planned and coordinated across sectors, and address the whole range of mutually-reinforced disparities that PwD have been experiencing during the COVID-19 pandemic and beyond. Indeed, our review also determined that lack of disability-inclusive response and emergency preparedness and pre-pandemic disparities created structural disadvantages which were further exacerbated during the pandemic. Both structural disparities and their pandemic ramifications need to be addressed by disability-inclusive public health and policy measures.”
Finally, I would suggest reviewing the references list, for example 8 and 15 are the same.
The duplicative reference was removed. The duplication arises from references in the Introduction that is also in the Results, while the EndNote file from the results was a different one – contains only the included papers. We carefully checked for other duplications of the same type (papers cited in the Introductions that were cumulatively inclusions in the review) and also found this one:
Jesus, T.S.; Landry, M.D.; Jacobs, K. A 'new normal' following COVID-19 and the economic crisis: Using systems thinking to identify challenges and opportunities in disability, telework, and rehabilitation. Work (Reading, Mass.) 2020, 67, 37-46, doi:10.3233/wor-203250.
The duplicative reference was removed in this case too.
Congratulations on this article and I hope it can bring more discussion about difficult life in pandemic times.
We appreciate it.

Reviewer 2 Report
Thank you for the opportunity to review this interesting study that provides an important contribution to the literature about PwD and the COVID pandemic.
There are some minor typos (eg line 1) that can be fixed with read through.
There does not seem to be mention of issues related to PPE despite widespread public reporting of supply and use issues, and associated planning/logistics failures. If it did not emerge in the literature included in the review there is likely a significant gap, particularly given the relative high risk faced by PwD. This is worth noting for further research and inclusion in the discussion.
Author Response
Response to the Reviewers
We would like to extend our gratitude to all the reviewers for the rapid and precise feedback, which helped us to improve the quality of the manuscript.
We respond below to each comment and highlight any resultant change in the manuscript. We do so in blue, while the reviewers’ comments are transcribed in black.
Reviewer 2
Thank you for the opportunity to review this interesting study that provides an important contribution to the literature about PwD and the COVID pandemic.
There are some minor typos (eg line 1) that can be fixed with read through.
Yes, these were fixed.
There does not seem to be mention of issues related to PPE despite widespread public reporting of supply and use issues, and associated planning/logistics failures. If it did not emerge in the literature included in the review there is likely a significant gap, particularly given the relative high risk faced by PwD. This is worth noting for further research and inclusion in the discussion.
The reviewer is correct that issues of Personal Protective Equipment (in terms of supply and logistics) did not come as prevalent as a form of a disability disparity in the literature. One likely reason is that this was not an issue specific for PwD. Even though it may have impacted on PwD, it may have been in the same proportion as the overall population. As it may not have been a ‘disproportional’ impact on PwD, i.e., not a disability disparity, the issue is not germane to the purposes of this paper in particular – although it may be a prevalent and important issue overall.
Finally, in a precedent paper (first review results from the whole scoping review project), we analyse factors such as the greater difficulties that some PwD may have in understanding or complying with the proper use of masks, handwashing, etc. This was framed in the literature as a disparity, but related with the use (and disability-accessible information on their use), not in terms of a disparity from failures in supply or logistics - and may have impacted the population overall. Hence, the issue was not found in the literature and not framed here as a lockdown-related disparity that could have a disproportional impact on PwD.

Reviewer 3 Report
Dear Authors,
the impact of the current pandemic is transversal to human nature itself, affecting health and well-being at all levels.
Structural racism, alongside discrimination and stigma are an atavistic problem, which has its roots in the foundation of some States and which only a purposeful and proactive cultural movement can overturn over the years.
People with Disabilities are among the categories most affected by these phenomena.
Paper needs a finishing touch and some insights in order to raise the bar of a paper, at a first glance already a notch above others.
Thank you for sharing your contribution proposal.
- why not SARS-CoV-2 among keywords? why not stigma/discrimination (not instead of, but alongside disparities, inequalities):
- please try to contextualize and deepen the current pandemic scenario when introducing your work;
- please refer to John Hopkins Coronavirus Resource Centre data instead of "Worldometer";
The PRISMA flowchart is not very clear, you have excluded works in the first phase (screening) and second phase (eligibility) according to the same criteria; explain this step better.
Honestly I would not have considered the pre-prints; although they are representations of an almost cleared phenomenon, they cannot be considered as scientific publications and, therefore, should be excluded from this work. Many pre-prints don't see the light. I consider it a limiting step.
Figure 2 is quite chaotic and hard to understand, try to make it more harmonious.
Your discussions are valid, even if can be improved further.
You need conclusions, too poor in this first attempt. What are the possible repercussions? What suggestions to give to the health policy maker? Define a clear "take home message" from your perspective and address a conclusion section. You need conclusions after such a work!
- in such a peculiar scenario, alongside socio-economic factors you should also deal with other factors in your discussion: what about risk perception, stigma and discrimination?! You have to deal with other psi-factors
You even didn't slightly refer to the very recent introduction of vaccines; deal with it, even because CoViD-19 vaccines are now available all over the globe (please also refer to https://www.who.int/news-room/spotlight/ten-threats-to-global-health-in-2019 )
Please update these gaps referring to the following references:
- Irigoyen-Camacho, M.E.; Velazquez-Alva, M.C.; Zepeda-Zepeda, M.A.; Cabrer-Rosales, M.F.; Lazarevich, I.; Castaño-Seiquer, A. Effect of Income Level and Perception of Susceptibility and Severity of COVID-19 on Stay-at-Home Preventive Behavior in a Group of Older Adults in Mexico City. Int. J. Environ. Res. Public Health 2020, 17, 7418
- Baldassarre, A.; Giorgi, G.; Alessio, F.; Lulli, L.G.; Arcangeli, G.; Mucci, N. Stigma and Discrimination (SAD) at the Time of the SARS-CoV-2 Pandemic. Int. J. Environ. Res. Public Health 2020, 17, 6341
- Sarah Dryhurst, Claudia R. Schneider, John Kerr, Alexandra L. J. Freeman, Gabriel Recchia, Anne Marthe van der Bles, David Spiegelhalter & Sander van der Linden (2020) Risk perceptions of COVID-19 around the world, Journal of Risk Research, DOI: 10.1080/13669877.2020.1758193
- Wong, B.Y.-M.; Lam, T.-H.; Lai, A.Y.-K.; Wang, M.P.; Ho, S.-Y. Perceived Benefits and Harms of the COVID-19 Pandemic on Family Well-Being and Their Sociodemographic Disparities in Hong Kong: A Cross-Sectional Study. International Journal of Environmental Research and Public Health 2021, 18, 1217
- Dye, T.D.; Alcantara, L.; Siddiqi, S.; Barbosu, M.; Sharma, S.; Panko, T.; Pressman, E. Risk of COVID-19-related bullying, harassment and stigma among healthcare workers: an analytical cross-sectional global study. BMJ Open 2020, 10, e046620
- Oliveira, Roberta Gondim de, Cunha, Ana Paula da, Gadelha, Ana Giselle dos Santos, Carpio, Christiane Goulart, Oliveira, Rachel Barros de, & Corrêa, Roseane Maria. (2020). Racial inequalities and death on the horizon: COVID-19 and structural racism. Cadernos de Saúde Pública, 36(9), e00150120. Epub September 18, 2020. https://dx.doi.org/10.1590/0102-311x00150120
- Gee GC, Ford CL. STRUCTURAL RACISM AND HEALTH INEQUITIES: Old Issues, New Directions. Du Bois Rev. 2011;8(1):115-132. doi:10.1017/S1742058X11000130
- Bailey ZD, Feldman JM, Bassett MT. How Structural Racism Works - Racist Policies as a Root Cause of U.S. Racial Health Inequities. N Engl J Med. 2021 Feb 25;384(8):768-773. doi: 10.1056/NEJMms2025396. Epub 2020 Dec 16. PMID: 33326717
Your contribution has no importance from an epidemiological and biostatic point of view, but it could be crucial for equality and human rights, offering a timely suggestion to the health authorities and policy makers.
Best regards,
Author Response
Response to the Reviewers
We would like to extend our gratitude to all the reviewers for the rapid and precise feedback, which helped us to improve the quality of the manuscript.
We respond below to each comment and highlight any resultant change in the manuscript. We do so in blue, while the reviewers’ comments are transcribed in black.
Reviewer 3
Dear Authors,
the impact of the current pandemic is transversal to human nature itself, affecting health and well-being at all levels. Structural racism, alongside discrimination and stigma are an atavistic problem, which has its roots in the foundation of some States and which only a purposeful and proactive cultural movement can overturn over the years. People with Disabilities are among the categories most affected by these phenomena.
We agree that PwD are among the most affected, and hence our motivation to develop this paper.
Paper needs a finishing touch and some insights in order to raise the bar of a paper, at a first glance already a notch above others. Thank you for sharing your contribution proposal.
Thank you, we respond to the specific suggestions below.
- why not SARS-CoV-2 among keywords? why not stigma/discrimination (not instead of, but alongside disparities, inequalities):
We added the SARS-CoV-2 as well as: healthcare disparities; social discrimination; social stigma; social inclusion. There are now 10 keywords (all MeSH terms in the PubMed database, for indexation purposes), which is the maximum allowed by the journal.
- please try to contextualize and deepen the current pandemic scenario when introducing your work; please refer to John Hopkins Coronavirus Resource Centre data instead of "Worldometer";
We now begin the Introduction as the follows (the resource suggested is now the reference # 1):
“The coronavirus disease 2019 (COVID-19), caused by the severe acute respiratory syndrome coronavirus 2 (SARS-CoV-2) strain, emerged in the late 2019, and since then has caused a global public health crisis of unusual proportions. Up to March 16, there were over 120 million cases and over 2.6 million deaths confirmed worldwide.[1] However, the impact of the COVID-19 pandemic, including the unintended effects of the measures to contain the pandemic (e.g., lockdowns), have not been equal across population groups.[2-4]
People with disabilities…”
The PRISMA flowchart is not very clear, you have excluded works in the first phase (screening) and second phase (eligibility) according to the same criteria; explain this step better.
It is a standard systematic/scoping review practice to use the same eligibility criteria for both the Level 1 and level 2 screening. The difference is that the criteria apply to the information on the titles-and-abstracts only (level 1), and then to the full texts (Level 2 assessment). When the inclusion/exclusion is not clear in the Level 1 (using the information from the abstracts alone), one retains the paper for full-text assessment and then includes or eliminates through the Level 2 screening (full-text assessment), using the same criteria but rather applied to the additional information that came from the full text. As this is the standard practice in roughly any review, and the published protocol paper outlines that we are going to use the same criteria for both stages, here we prefer not to specifically report on that standard process. We are concerned about the length of the paper, and this and other similar reports of standard methodological practices (that were reported in the previously published study protocol) would increase the paper length substantially, possibly without a need.
Honestly I would not have considered the pre-prints; although they are representations of an almost cleared phenomenon, they cannot be considered as scientific publications and, therefore, should be excluded from this work. Many pre-prints don't see the light. I consider it a limiting step.
We agree that preprints cannot be considered scientific publications and are not equivalent to peer-reviewed literature. In our eligibility criteria, Methods section, we report that: “[we] included any peer-reviewed empirical, conceptual or perspective paper (e.g. editorials, commentaries) or preprint empirical studies”. This highlights that they are not considered to be of the same kind.
In the published and open-access study protocol, we detailed reasons for including this literature, but we did not report that in this paper. As this is not a standard feature, we understand we should do it here, as well. Hence, briefly, in the methods subsection 2.2 of this paper, we now report the following (new content underlined):
“[…] preprint servers (i.e., MedRxiv, SocArXiv, and PsyArXiv) were searched for to identify empirical or perspective papers meeting the eligibility criteria. During the initial stages of the COVID-19 pandemic, preprint databases have been hosting many studies that have not reached peer-reviewed publications yet.[37] This option can also help to avoid the exclusion of studies with negative results which may be published less often or less rapidly.”
Newly cited reference - number 37:
Gianola, S.; Jesus, T.S.; Bargeri, S.; Castellini, G. Characteristics of academic publications, preprints, and registered clinical trials on the COVID-19 pandemic. PloS one 2020, 15, e0240123, doi:10.1371/journal.pone.0240123.
Also, we missed to report the following step that we took, which is relevant for the case. Hence, immediately after the content above, we now also mention that:
“Before the data charting, we searched for the peer-reviewed version of the included preprints, and have replaced the record whenever found.”
Perhaps more importantly, the inclusion of preprints (reaching peer-reviewed publication or not later on) could have been considered an important limitation of a systematic review, depending on a number of factors. However, this is a scoping review, and scoping reviews can - and often should - include a wider range of literature, which is not typical in systematic reviews. These can include policy papers, legal papers, websites, or even blogs - as explicitly articulated by the methodological guidance on scoping reviews.
Please see to the recent guidelines for the conduct of scoping reviews, updated in 2021 from an initial version in 2015, for further references:
Peters MD, Godfrey CM, Khalil H, McInerney P, Parker D, Soares CB. Guidance for conducting systematic scoping reviews. International journal of evidence-based healthcare 2015, 13(3):141-146.
Peters MDJ, Marnie C, Tricco AC, et al. Updated methodological guidance for the conduct of scoping reviews. JBI evidence synthesis. JBI Evid Implement. 2021, 19(1):3-10.
This context notwithstanding, we agree we can and should make a note in the Limitations about the need to carefully interpret finding from preprint studies (new content underlined).
“perspective [papers) should be carefully interpreted. The same applies to the preprint studies, which were not peer-reviewed; therefore, their findings should be interpreted with caution, as well.”
Figure 2 is quite chaotic and hard to understand, try to make it more harmonious.
Our figure and overall results are aimed at “synthesizing the complex net of health, socio-economic or participation disparities experienced by PwD as a result from lockdown-related measures”, as we mention in the synthesis subsection of the Methods. More importantly, when we first report to the Figure in the results, we note that:
“Figure 2 provides a graphic display of our themes. The lockdown-related disparities were initially organized in 10 primary themes, with complex inter-links as well as often-blurred limits among one another. Our findings suggest that these 10 disparities are ramifications that stem from 2 underlying factors, here framed as central or overarching themes.”
The figure highlights the 10 ramifications that stem from 2 underlying, intricate causes. If some chaos in reported the figure, it is something that we are willing to retain – not to oversimplify the representation of the complex, intricate phenomenon. The figure tries to portray the complexity, which exists, for example, in representations of the theory of complex adaptive systems.
Finally, I will supply a full-page figure, which adds to the spacing of the elements within, and may contribute to an easier grasp. We have also made the text in the revised figure bold
Your discussions are valid, even if can be improved further. You need conclusions, too poor in this first attempt. What are the possible repercussions? What suggestions to give to the health policy maker? Define a clear "take home message" from your perspective and address a conclusion section. You need conclusions after such a work!
We agree we need to provide some more in the conclusions, in line with suggestions detailed in the discussion. For example, we need to outline the need for policy-makers to develop coordinated responses across sectors, as the disability disparities synthesized pertain to multiple sectors and relate in complex ways with one another.
Perhaps more importantly, we now inform the reader that a subsequent paper (last results from this whole scoping review project) will explicitly synthesize recommendations for action – not from ourselves, but the ones we have extracted from the reviewed literature itself – that will more specifically guide policy-makers and other stakeholders on the what and how to develop disability-inclusive responses. Previously, we failed to report this critical piece of information. The interested reader won’t be empty handed with regards to a blueprint for action, even though we cannot cite the upcoming reference beforehand.
Our conclusion now reads as it follows (new content underlined):
“Lockdown-related measures to contain the COVID-19 pandemic can disproportionally affect PwD in health, educational, social support, social participation, and socio-economic terms. These disparities influenced one another, and arguably so as public health inequities and occupational injustices are often determined by social determinants of health and occupation [36,37,125]. Hence, public health and policy interventions, including social policies, might be planned and coordinated across sectors, and address the whole range of mutually-reinforced disparities that PwD have been experiencing during the COVID-19 pandemic and beyond. Indeed, our review also determined that lack of disability-inclusive response and emergency preparedness and pre-pandemic disparities created structural disadvantages, which were further exacerbated during the pandemic. Both structural disparities and their pandemic ramifications need to be addressed by disability-inclusive public health and policy measures. We will detail specific actions to address these disparities in the last results paper (upcoming) of this whole scoping review project. Such a paper will synthesize the actions or recommended actions that the literature has reported on how to address the full range disability disparities related with the COVID-19 pandemic, which were identified both here and elsewhere[1].”
- Your discussion and conclusion caress a hypothesis concerning social inequalities, but you must deepen this point. SES is a crucial point when attempting to interpret the data. Please feel free to refer to a milestone such as National Research Council (US) Panel on Race, Ethnicity, and Health in Later Life; Anderson NB, Bulatao RA, Cohen B, editors. Critical Perspectives on Racial and Ethnic Differences in Health in Late Life. Washington (DC): National Academies Press (US); 2004. 9, Race/Ethnicity, Socioeconomic Status, and Health and also Williams DR, Priest N, Anderson NB. Understanding associations among race, socioeconomic status, and health: Patterns and prospects. Health Psychol. 2016;35(4):407-411. doi:10.1037/hea0000242
It is true that our discussion raises the point of social inequalities. Racial and ethnic disparities are important with these regards, and these have been important issues during the pandemic, as well. However, we prefer to retain our focus exclusively on the inequalities experienced by PwD. This is the population we address of this scoping review in particular.
2. in such a peculiar scenario, alongside socio-economic factors you should also deal with other factors in your discussion: what about risk perception, stigma and discrimination?! You have to deal with other psi-factors
We agree with the review that factors such as stigma and discrimination were under-addressed in our review. Therefore, we now include the following content in our discussion (new content underlined):
“From the central themes in our analysis, it is suggested that lockdown-related disparities experienced by PwD often arise from structural disparities that PwD were experiencing before the pandemic, and were exacerbated thereafter. In turn, the structural disparities often arise from stigma, ableism, discrimination, and marginalization of PwD - still prevalent in societies, which lead to the social exclusion of PwD and have contributed to the diverse disability disparities observed during the COVID-19 pandemic.[15,77,90,126]. Hence, fundamentally addressing the disability stigma and discrimination in societies can contribute to address the structural determinants of disparities experienced by PwD, during pandemics and beyond.”
The references cited here are these ones:
Andrews, E.E.; Ayers, K.B.; Brown, K.S.; Dunn, D.S.; Pilarski, C.R. No body is expendable: Medical rationing and disability justice during the COVID-19 pandemic. The American psychologist 2020, doi:10.1037/amp0000709.
Lund, E.M.; Forber-Pratt, A.J.; Wilson, C.; Mona, L.R. The COVID-19 pandemic, stress, and trauma in the disability community: A call to action. Rehabilitation psychology 2020, doi:10.1037/rep0000368.
D'Cruz, M.; Banerjee, D. 'An invisible human rights crisis': The marginalization of older adults during the COVID-19 pandemic - An advocacy review. Psychiatry research 2020, 292, 113369, doi:10.1016/j.psychres.2020.113369.
Sabatello, M.; Burke, T.B.; McDonald, K.E.; Appelbaum, P.S. Disability, Ethics, and Health Care in the COVID-19 Pandemic. American journal of public health 2020, 110, 1523-1527, doi:10.2105/ajph.2020.305837.
You even didn't slightly refer to the very recent introduction of vaccines; deal with it, even because CoViD-19 vaccines are now available all over the globe (please also refer to https://www.who.int/news-room/spotlight/ten-threats-to-global-health-in-2019 )
We did not address the issue of vaccines, in this paper in particular, for some reasons:
On the one hand, the literature addresses the first wave of the COVID-19 pandemic. This was reflected inclusively in the title of this paper. Issues of vaccines and any disparities in their distribution relative to PwD was not an issue in the literature, at least by that timing. On the other hand, this paper focuses on lockdown-related disparities - with a broader health and social scope. The previous review results from this whole scoping review project, which focused directly on health risks and consequences of COVID-19 infection, slightly addressed issues related to vaccination of PwD (e.g. of people with down syndrome who usually have lower immune responses). Moreover, the final, upcoming paper of this scoping review project (on a model action to address health and lockdown-related disability disparities) will explicitly discuss how that model can be inclusive of current vaccination issues, using references of the WHO and others too.
All and all, while we agree that the overall issue of vaccination is clearly are relevant, we understand that the issue has a more specific application in the upcoming paper - specifically focused on a blueprint for further action to take.
Please update these gaps referring to the following references:
- Irigoyen-Camacho, M.E.; Velazquez-Alva, M.C.; Zepeda-Zepeda, M.A.; Cabrer-Rosales, M.F.; Lazarevich, I.; Castaño-Seiquer, A. Effect of Income Level and Perception of Susceptibility and Severity of COVID-19 on Stay-at-Home Preventive Behavior in a Group of Older Adults in Mexico City. Int. J. Environ. Res. Public Health 2020, 17, 7418
- Baldassarre, A.; Giorgi, G.; Alessio, F.; Lulli, L.G.; Arcangeli, G.; Mucci, N. Stigma and Discrimination (SAD) at the Time of the SARS-CoV-2 Pandemic. Int. J. Environ. Res. Public Health 2020, 17, 6341
- Sarah Dryhurst, Claudia R. Schneider, John Kerr, Alexandra L. J. Freeman, Gabriel Recchia, Anne Marthe van der Bles, David Spiegelhalter & Sander van der Linden (2020) Risk perceptions of COVID-19 around the world, Journal of Risk Research, DOI: 10.1080/13669877.2020.1758193
- Wong, B.Y.-M.; Lam, T.-H.; Lai, A.Y.-K.; Wang, M.P.; Ho, S.-Y. Perceived Benefits and Harms of the COVID-19 Pandemic on Family Well-Being and Their Sociodemographic Disparities in Hong Kong: A Cross-Sectional Study. International Journal of Environmental Research and Public Health 2021, 18, 1217
- Dye, T.D.; Alcantara, L.; Siddiqi, S.; Barbosu, M.; Sharma, S.; Panko, T.; Pressman, E. Risk of COVID-19-related bullying, harassment and stigma among healthcare workers: an analytical cross-sectional global study. BMJ Open 2020, 10, e046620
- Oliveira, Roberta Gondim de, Cunha, Ana Paula da, Gadelha, Ana Giselle dos Santos, Carpio, Christiane Goulart, Oliveira, Rachel Barros de, & Corrêa, Roseane Maria. (2020). Racial inequalities and death on the horizon: COVID-19 and structural racism. Cadernos de Saúde Pública, 36(9), e00150120. Epub September 18, 2020. https://dx.doi.org/10.1590/0102-311x00150120
- Gee GC, Ford CL. STRUCTURAL RACISM AND HEALTH INEQUITIES: Old Issues, New Directions. Du Bois Rev. 2011;8(1):115-132. doi:10.1017/S1742058X11000130
- Bailey ZD, Feldman JM, Bassett MT. How Structural Racism Works - Racist Policies as a Root Cause of U.S. Racial Health Inequities. N Engl J Med. 2021 Feb 25;384(8):768-773. doi: 10.1056/NEJMms2025396. Epub 2020 Dec 16. PMID: 33326717
We found the references above interesting and relevant pieces of literature.
For example, the second one addresses the issues of discrimination and stigma we now address in our discussion (see above). However, even that reference was not specific or directly focused on the stigma and discrimination that PwD experience. In contrast, we had - among the 127 references that we do cite - specific references that outline the role of stigma, discrimination, and disability-specific social factors (ableism) on disability disparities. Hence, we use four references (detailed above in our response to the ‘stigma and discrimination’ issue) that we have for that role.
We also found very interesting, for example, the N Engl J Med’s reference on structural racism. But once again, the issue only could indirectly relate with PwD. The same applied to all the other references suggested, when the great majority of the 127 references we that cite are specific to the population we address: PwD.
Your contribution has no importance from an epidemiological and biostatic point of view, but it could be crucial for equality and human rights, offering a timely suggestion to the health authorities and policy makers.
We are delighted to read that, as that was really our intention. Thank you for the overall review and this remark in particular.
Additionally, we took the initiative to replace the acronym PwD and we have used rather the full abbreviation as “people with disabilities” in the revised manuscript file.

Round 2
Reviewer 3 Report
Well done
Author Response
We are grateful for the Editor's constructive feedback, helpful suggestions, and insightful comments. We appreciate the quality review and careful attention to our paper and believe that suggested changes have enhanced and improved the quality of the manuscript.
As requested, we addressed each comment below (in blue) to the best of our ability and highlighted the corresponding changes in the manuscript (using Track Changes).
Dear Dr. Jesus,
Thank you for your careful attention to the reviewers' comments. At this round of the review process, we have several comments, that we believe would enhance the readability and usefulness of the manuscript. While some are of a purely editorial nature some will require more extensive revision, and therefore we are asking that you do major revisions before resubmitting the manuscript. Please also prepare a detailed point-by-point response to all points mentioned below:
1. Web appendix 1: This level of detail is not necessary. What is needed is a table that provides an overview of the studies included with all necessary information for each study in this one table. What we propose is this: a) add three more columns: one with the geographical focus, one with the population, and one with the publication type for each publication; b) substitute the first column (codes) for the authors' names and the publication date; c) have only one box summarising the main findings, rather than showing separately each reviewer; and d) remove the two columns with the data extraction (the two columns starting PwD).
In response to the Editor’s suggestions we have made the following changes:
- added the three additional columns, as requested;
- substituted the first column (codes) for the authors' names and the publication date;
- merged the content from both reviewers, as requested;
- merged the data extraction from both reviewers, as done for the request c). In the other paper submitted to this journal from this review project, we were given the option to either remove these two columns with data extractions or clarify what it means and streamline its report. We followed the latter and would like, for the same reasons and consistency, to follow the same approach here.
Hence, in doing so, we first added the following heading over these two columns:
“Textual Data Extractions - all the extractions are direct quotations from the articles”.
Also, we further harmonized the reporting approach, which previously had quotation marks used inconsistently. With the new heading, we now could remove all the quotation marks, as it is clear that all the content in these columns refers to quotations from the articles.
- In the keywords, “disabled persons” can be replaced by “people with disabilities”
All the keywords in this manuscript were MeSH terms, given that (a) it is important for the correct indexation in databases for PubMed, and (b) many health journals require using MeSH terms as keywords. While we were not sure about the policies of this journal, we followed the Editor’s suggestion and made a replacement.
- In the keywords, “social discrimination” can be “discrimination” and “social stigma” can be “stigma”.
The same issue as above (p.2). We replaced the keywords according to the Editor’s suggestion.
- In the Background chapter, the definition of “persons with disabilities” is introduced. In the UN CRPD, the definition is a bit different in terms of long-term nature: "Persons with disabilities include those who have long-term physical, mental, intellectual or sensory impairments which in interaction with various barriers may hinder their full and effective participation in society on an equal basis with others.” Would the authors like to reflect on this definition?
We appreciate the suggestion and the opportunity to reflect upon and refine the definition of persons with disabilities. While we are aware of the UN CRPD definition (2006), we also recognize that this type of exclusive focus on the long-term nature has been challenged more recently. The understanding and definitions of disability increasingly address the notion of experience that most people had or will have at some point in time, at least by some period of time. See for example one of the references we cite at the point we introduce the term - from the disability and rehab leader of the WHO:
Cieza, A.; Sabariego, C.; Bickenbach, J.; Chatterji, S. Rethinking Disability. BMC medicine 2018, 16, 14, doi:10.1186/s12916-017-1002-.
We are more aligned with the more recent perspectives, which we cite, and are not, in this work, exclusively focused on the long-term nature of the disability.
We also recognize that the issue of “an equal basis with others”, addressed by the UN CRPD’s definition and others, was only implicit and that should be explicit. Hence, it is now stated that:
“People with disabilities include people who experience, at any given point in their lifespan, any mobility, intellectual, cognitive, developmental, or sensorial impairments which in interaction with environmental factors may hinder their daily functioning and social participation on an equal basis with others”
- In the background chapter, at the end of the second paragraph, "these forms health disparities” —> “these forms of health disparities”?
Corrected. Thank you.
- As for the vulnerability, please be more careful in using the concept and avoid “labeling” people with disabilities as inherently vulnerable.
Agreed. New content underlined:
“Additionally, people with disabilities are historically a socially vulnerable (not inherently vulnerable)…”
- On page 3, 3rd line: "quarantines, of People” —> “quarantines of people”. At the end of the same line, there is perhaps “People with disabilities” that should be erased before the sentence in line 4 starting from “Finally,”
Corrected.
8. In the “Methods” chapter, the first line of “methodology” is perhaps “methods”?
There were further changes here as a result of another suggestion. Please see our response and resultant changes at point #25.
- Figure 2 has relatively little color contrast. I would recommend the authors select a background that allows enough color contrast to be more accessible for readers with visual disabilities.
We revised Figure 2 to increase/maximize contrast. We replaced the text labels with textboxes featuring bold black font on white background.
- Section 3.2 enlists summaries of different studies. More analysis can be conducted on the findings of the systematic review to increase the readability to go beyond the list of summaries. Are there any relationships among different sub-themes, for instance?
Section 3.2 provides an aggregative thematic analysis of the combined papers. There are instances in which more than ten papers contributed to one statement, which does not equate, in our view, to enlisted summaries of different studies - even though it is true that at times one reference was key or unique for a given statement. Moreover, the same study (i.e., different extractions from different contents of it) is often cited in many different themes throughout the results sections. This section not merely narratively lists what each paper has found. Instead, it describes 10 primary themes and 2 overarching themes, with contents from each paper that applied and were used to build these themes being described and cited, sometimes multiple times each one, as appropriate.
Yes, there are multiple relationships among themes and sub-themes, which we address throughout the paper (pg. 7, 8, 9, 11, 12-15, 18, 20).
For example, when introducing Figure 2 (pg. 7), we say that:
“The lockdown-related disparities were initially organized in 10 primary themes, with complex inter-links as well as often blurred limits among one another. Our findings suggest that these 10 disparities are ramifications that stem from 2 underlying factors, here framed as central or overarching themes.”
Then, in the next paragraph (pg. 7), we say:
“Some of the primary themes (#1 to #5) address specific disparity types, while some others (#6 to #8) address consequences from the combinations of these disparities. One theme (#9) focuses on the vicious circle of additional vulnerabilities (e.g., reduced healthcare access) arising from reduced employment and income. A tenth theme addresses the inaccessibility of tele-solutions for many people with disabilities, which could partly compensate for the other disparities. Finally, all these themes stem from 2 central themes: the lack of disability-inclusive responses and emergency preparedness, and the structural, socially-entrenched disadvantages people with disabilities were experiencing before the pandemic.”
Then, for example, in theme # 1, we report the following direct links to other themes (pg. 8):
“People with disabilities had reduced access to in-person healthcare services due to lack of appropriate transportation during lockdowns. [49,71,116] Access to alternative telehealth solutions was hampered too (details in theme #10), while financial coverage for healthcare services also can be reduced (details in theme #9).”
In theme #5 (pg. 11), we reported a link to theme four, now reinforced (new content underlined):
“As school services often provide a reliable source of meals, learning opportunities, social participation, and may serve as supportive environments for the most vulnerable children with disabilities (e.g., young refugees, young migrants, children living without parental care, homeless children, children living in urban slums, children in conflict-affected areas), the closures of schools and inherent support systems (see theme #4) risks bringing the greatest drawbacks and aggravate existing social and educational disparities for many children with disabilities.[30,67]”
Theme #6 begins as follows (pg. 11):
“People with disabilities can be especially vulnerable to numerous psychological consequences of disrupted routines, activities, and support networks (see aforementioned).”
There are other examples across the results section (e.g., the family and informal caregiving stress (theme #7) magnified by the disruption of services/support networks (theme # 4), the risk of maltreatment and reliance in the perpetrator, at the end of theme # 8, explicitly linked to the financial struggles addressed in the themes # 9). [pg. 12-15]
To further complement the points, and an additional means to illustrative the multiple and often intricate relationships that occur between both, themes and sub-themes, in the Discussion, we specifically dedicate an entire paragraph to that, with a direct implication being stated (pg. 18):
“The distinct themes and sub-themes identified and presented through this analysis are intricate and interconnected. For example, added risks to maltreatment, negligence, or abuse toward people with disabilities arise due to service closures, across a range of the health, educational and social sectors. These closures removed key supportive services and part of the societal control over maltreatment, which could not be replaced by digital solutions that are not designed to be used independently by people with disabilities. In turn, the disruption of extended family or community support networks also contributed to an increased caregiving burden, which can turn maltreatment more likely. Finally, harder economic conditions can also lead to increased family stress and a greater reliance of people with disabilities on the perpetrator, within the household. These factors, addressed within multiple themes, illustrate how these can be interdependent.
Indeed, in this context of multiple, mutual, and intricate relationships, effective policy responses cannot be fragmented, e.g., need to be intersectoral, to address, at the same time, the whole set of factors that contribute to disability disparities.”
Finally, also in the conclusions (pg. 20), we state that:
“Lockdown-related measures to contain the COVID-19 pandemic can disproportionally affect people with disabilities in health, educational, social support, social participation, and socio-economic terms. These disparities influenced one another, and arguably so as public health inequities and occupational injustices are often determined by social determinants of health and occupation[34,35,124]. Hence, public health and policy interventions, including social policies, might be planned and coordinated across sectors, and address the whole range of mutually-reinforced disparities that people with disabilities have been experiencing during the COVID-19 pandemic and beyond.”
- On page 13, “their loved ones” can be replaced by “their family members with disabilities”
It is now replaced.
- On page 13, references to 49 are separate from other references (30,51,67,75). 49 should also be part of them?
Thank you. It is now corrected.
- on page 14 and 16, “patients” should be replaced by “people”
Replaced.
- on page 14, in the last sentence, the sentence begins with “,” which should be placed at the end of the previous line.
Corrected.
- I wonder how “frail older adults” are defined in this paper. Frailty is not necessarily a disability, but this paper gives the impression that it is part of the disability. Please kindly clarify the relationship between frailty and disability.
We thank the Editor for this insightful comment.
Our working definition of people with disability to inform eligibility decisions – in the study protocol and now also here, in the new Appendix 1, stated the following:
“For this study, “people with disability” are defined as those experiencing, at any point across their lifespan, long- or short-term impairments in 1 or more body structures or functions (e.g., affecting mobility, sensorial, intellectual, communication, or cognitive function) arising from a health condition or natural processes (e.g., aging) which, in interaction with environmental factors, affect the performance of daily activities or social participation.”
The issue of frail elders is not fully clarified by the above, although the issue of ‘aging’ above allowed for further consideration. During our reviewers’ discussion on eligibility decisions, a posteriori, we needed to set more clear-cut criteria on this. For example, articles addressing older adults or frail elders with no further, explicit link to functional limitations in the addressed population were excluded. Yet, if it was explicit in the full text that the ‘frail elders’ also had functional limitations in ADLs, social participation, etc., the article was included.
To avoid confusion, the terms ‘frail’ or ‘frail elders’, which in fact here mean frail elders that cumulatively experience functional limitations, were replaced throughout the article, for example by ‘older adults experiencing disabilities’ (e.g., pg. 9, 14).
- On page 19, “people with disabilities” are three times repeated. Please erase the phrase.
Thank you for catching this accidental repetition. We are not sure how this happened – it is now deleted.
- The discussion and conclusion chapters are to some extent similar to the other scoping review paper you submitted on the topic in this journal- please ensure there is specificity.
We find it inevitable and unavoidable that, “to some extent”, some issues were discussed in various parts of discussion chapters of both papers. For example, a great part of the manuscript’s methods derives from a common trunk (a whole scoping review project), hence, some of the methods-based limitations of one paper are also present in the other. Given that the papers should be self-sufficient – which, for example, may have led to the instructions we received to provide information that was previously published in the study protocol, there is little room to avoid, in these points, some degree of similar content. By the same token, the limited number of empirical studies and especially the small number of papers focused on LMICs were examples of some issues that also were common in both review results, hence commonly addressed. But once again, we don’t see another way to reconcile this issue at this point and to maintain the papers self-sufficient at the same time.
On the other hand, we recognize that the six first paragraphs of this discussion are entirely focused on lockdown-related disparities, which are the results specific only for this paper. While the paragraphs of the discussion of the previous paper mainly addressed medical triage issues, the related unethical disadvantages, multiple infection risks for residents of long-term facilities, and lack of accessible information for people with disabilities on COVID-19 risks and control measures. Thus, we believe that the discussions are not very much similar in scope or content.
We also reviewed the conclusion, and the same issue seems to apply. Below, we transcribe, respectively, the conclusion from the previous paper and that of the current one. The first conclusion is focused on disparities related to a COVID-19 infection, while the second is focused on the disparities related to lockdown-related measures – which are distinct subjects that, although partly related, can be set apart. In response to the Editor’s suggestion, in the conclusion of this paper (the second transcription), we have made some changes (underlined) to clarify and reinforce we are specifically referring to lockdown-related disparities in the subject of the sentences. We also have eliminated the last sentence that linked to a forthcoming paper (currently under review) – per an upcoming recommendation we totally agree with (this also responds to the last point of this review: #32).
Conclusion of the other paper:
“This scoping review, addressing the initial stages of the COVID-19 pandemic, suggests that people with disabilities can experience disproportionate health risks and consequences from a COVID-19 infection. These risks and disparities are challenging and go beyond any health-related vulnerabilities (e.g., the greater presence of comorbidities among younger people with disabilities relative to non-disabled counterparts), and in fact entail multiple environmental factors. These include, for example, multiple exposure risks for People with disabilities who live in residential facilities, lack of accessible healthcare and information, and medical rationing affected by ableism and disability stigma. These environmental determinants of disparities can, and should, be modified to prevent or mitigate any disproportionate health risks and consequences of a COVID-19 infection for People with disabilities worldwide.”
Conclusion of this paper:
“Lockdown-related measures to contain the COVID-19 pandemic can disproportionally affect people with disabilities in health, educational, social support, social participation, and socio-economic terms. These disparities influenced one another, and arguably so as public health inequities and occupational injustices are often determined by social determinants of health and occupation [34,35,124]. Hence, public health and policy interventions, including social policies, might be planned and coordinated across sectors, and address the whole range of mutually reinforced, lockdown-related disparities that people with disabilities have been experiencing during the COVID-19 pandemic and beyond. Indeed, this review of lockdown-related disparities also determined that lack of disability-inclusive response and emergency preparedness, as well as those pre-pandemic disparities, created structural disadvantages which were further exacerbated during the pandemic. Both structural disparities and their pandemic ramifications need to be addressed by disability-inclusive public health and policy measures.”<previous last sentence removed>
- PwD should be spelled out as people with disabilities throughout the manuscript, figures, and appendices.
This has been done. Thank you.
- PRISMA
A) Merge the stages 'Full-text articles assessed for eligibility and 'Papers included in the scoping review project', and also their respective boxes with excluded articles. There should be one box titled 'full articles assessed for eligibility, 68 of which were excluded, with reasons.
B) The last box currently titled 'Papers included in the thematic analysis' needs to be renamed 'Paper included in the coping review'
We thank the Editor for this constructive suggestion. The above sections are now revised as instructed. Of note, as we were also reviewing the other paper submitted to the journal, we have introduced this structural change in the other paper as well, for consistency, even though it was not specifically requested for that paper.
- 'People with disabilities': the word 'people' should only be capitalized after a full stop. In all other instances use a small 'p'.
Corrected, throughout the paper.
- There are a few typographical errors and a couple of times you have pasted 'people with disabilities' in the text, where it is not needed- e.g. top of page 19.
We really appreciate your careful attention to our manuscript. We deleted the duplication. Also, there were a couple of instances where “disabilities’s” were replaced by “disabilities’” (pg. 15, 16).
- Remove Web appendix 2.
Removed.
- In the abstract you use vulnerabilities and disparities interchangeably; please avoid doing that. In most cases, although not always, 'disparities' are more appropriate.
We agree with this astute comment. Several changes have been made in the abstract (pg. 1) and the text (pg. 7, 14).
- Background: towards the end of this section you mention 'occupational injustice'; it is not clear what this refers to and needs to be removed. This term, coming from occupational science, does not seem to add to your argument.
Removed.
- Methods: This is a scoping review- it is not scoping review AND thematic analysis. Identify the study as a scoping review and in the analysis section discuss the hematic analysis method you used. Also, reading section 2.5 it seems that what you did as a narrative synthesis, bringing together data from a variety of sources, rather than a thematic analysis, which would require all studies to have been qualitative.
We used scoping review as the overall study method, utilizing the thematic analysis as the analytical approach (similar to a systematic review [overall method] with a meta-analysis [analytical approach], which is traditionally identified as a systematic review with meta-analysis).
We recognize that the scoping reviews can be conducted without thematic analysis and that the second is not implied in the first. We also agree that in the title we should avoid using “and”, which can be ambiguous, and rather use “with”. Additionally, we replaced “thematic synthesis” with “thematic analysis” which is more appropriate and accurate terminology.
The revised title of the article now reads:
“Lockdown-related disparities experienced by people with disabilities during the first wave of the COVID-19 pandemic: scoping review with thematic analysis.”
We also have made similar changes in the Abstract (pg. 1), and at the beginning of the Methods (pg. 3). The latter now reads:
“This paper uses a scoping review method with a thematic analysis as the analytical approach.”
We realize that in section 2.5, we did not provide specific details on the integrative, data-based convergence synthesis approach that we used to support the application of a thematic analysis – or any other approach to analyze or synthesize qualitative information. This approach is one of those that can be used in mixed-methods research or reviews (please see the corresponding references #43 and 44 in the revised manuscript). In this approach, unlike others (in which qualitative and quantitative data are analyzed and then reported completely apart) the different types of data are analyzed under the same method. More often, quantitative data is not combined or aggregated on their numerical or face value but analyzed ‘qualitatively’, considering their meanings or implications, alongside the actual qualitative data (and the published perspective papers as well). Under this approach, one type of data can be combined within the other under the same method and themes, and even help to clarify one another. To correct this omission in section 2.5, we now report the following (pg. 5):
“To enable this type of qualitative synthesis, out of mixed-methods data coming from the scoping review, we applied an integrative, data-based convergent synthesis approach, [43, 44] where qualitative and quantitative evidence, as well as published perspectives, were analyzed using the same synthesis method. In this case, all forms of data were analyzed qualitatively and together under the same themes, provided that they address the same contents. Specifically, quantitative data were not numerically aggregated with data from other quantitative papers included, but analyzed in their meaning and extracted implications alongside qualitative data and published perspectives within the same themes.[43,44] In short, these different types of information, were qualitatively combined within the same themes, in complement to (e.g., contributing to the interpretation of) one another.”
Other approaches to analyzed mixed-methods data analyze and present qualitative data on the one side - as completely separate entities, even if qualitatively addressing the same issue. We understood that using the same analytical approach, and one qualitative, for the different types of information would be more appropriate here. We do not claim it is the best or single approach, just the one we followed among the options provided by the cited references, provided that we cite the methodological sources (as we did before) and clarify the rationale (as we hopefully do now in the transcription above).
Finally, although at times, within the themes, we describe the methodology and specify what was found in one study or some of them within a storyline, overall we believe we have developed a thematic analysis above and beyond what could be classified as a narrative synthesis. In subsection 2.5 we report (pg. 5):
“We have developed a new interpretive schema and configuration, inclusive of both primary and central, overarching themes.”
In the upcoming comment #29, it was suggested that the central themes should be placed elsewhere (out of the results section). Given that (a) this feature is the key component of our interpretative schema (with overarching themes) and (b) the reflexive nature it entails, we argue that having these themes presented within the results is aligned with (or a requirement of) a reflexive thematic analysis approach*.
* Braun, V.; Clarke, V. Reflecting on reflexive thematic analysis. Qualitative Research in Sport, Exercise and Health 2019, 11, 589-597, doi:10.1080/2159676X.2019.1628806
- Definitions of people with disabilities and of vulnerability need to be included in this paper- it is not enough to refer to the published protocol.
We agree, and we now provide this content as a web appendix, now labeled as Web-appendix 1. This content, especially for the issues of vulnerability, has several pages, hence it is now presented in a web appendix.
- Table 1: subheadings seem to be missing
Thank you for catching this omission. Table 1 subheadings are now included (pg. 6).
- Details of the information sources and search process need to be given in this article- it is not enough to refer to the protocol.
We agree with this comment. The full search strategies for each database are now included as a sub-section of the new Web-appendix 1. We used subsections of a single appendix for all the new additions to avoid generating multiple appendixes.
- Section 3.3.: these two central themes contain information that would be better suited in the introduction and/or the discussion, but not in the findings- they give background information (introduction) and interpret the findings (discussion) and thus we suggest you eliminate section 3.3. and move the information to the background and the introduction, as appropriate, and ensuring there is no duplication of information.
This suggestion was previously evoked, as it was relevant to our response to point #25 (please see pg. 11 of this document). We believe that the central themes are best suited for presentation within the Results section, as this presentation follows the natural thematic flow and improves the understanding and readability of the manuscript. We would like to kindly ask the Editors to reconsider this suggestion. The issues in section 3.3 indeed emerged from the literature reviewed and papers included – i.e., from the cited papers. On the one hand, the issues with citations come from included papers and extracted material, hence can or should be rather part of the review Results. On the other hand, in a thematic analysis there is also a certain level of interpretation and reflection over the extracted content, either semantically present or not, and the need to develop a new interpretative schema, which here refers to the development of two, central themes. This kind of interpretation is promoted, if not required, in a thematic analysis – as previously mentioned. We believe that while there are other analytical approaches in which the analyses, interpretation, and discussion would be differently organized, these do not represent the most optimal approach for a thematic analysis, which we proposed (in the study protocol) and followed in this review.
- This sentence is unclear' These could prevent or mitigate the net of the disparities faced by people with disabilities during the initial lockdowns, should they exist.'
We agree that the sentence sounds vague. Moreover, it has no specific role in the context of our discussion; hence, it was deleted.
- Limitations: remove the sentence 'The last work from this scoping review project will focus on that.' And, more generally, remove all instances where you refer to not-yet-published studies- this manuscript must act independently from other manuscripts and should contain all necessary information.
Removed, as requested.
- Remove the last sentence: 'We will detail specific actions to address these disparities in the last results paper (upcoming) of this whole scoping review project. Such a paper will synthesize the actions or recommended actions that the literature has reported on how to address the full range of disability disparities related to the COVID-19 pandemic, which were identified both here and elsewhere [8].'
Removed, as requested.
Once again, we would like to thank the Editors for the high-quality review and careful attention to our manuscript.
JOURNAL’S EDITOR – ADDITIONAL COMMENTS
The references in the supplementary will not be taken into account as
being cited when the paper gets published, so we kindly ask you to move them
into the main text and cite them in the back matter of the paper, in the
"Supplementary Files" section.
The Supplementary (i.e. Web) Appendix that merely contained a list of references was eliminated per the request of the guest editor(s) below. Any other citations in the remaining or newly introduced appendixes were cited in the main text too. Hence, they are part of the references list in the main paper.
"Appendix" should be named "Supplementary" as it is not directly part of
the main article file.
Thank you. We proceeded accordingly
- Please make sure that all references are in the MDPI style.
(https://www.mdpi.com/authors/references)
We have used ENDNOTE’s MDPI output style to manage the references. There were a few cases in which the in-text citations were not together as they should e.g. [“X”] [“R-Z2]. These were now merged.
- Please decide upon a title as we are not allowed to use "running title".
We are happy to eliminate the running title and keep the full title.